# LINEAR DIFFUSION MODELS MEET CONTEXTUAL BANDITS WITH LARGE ACTION SPACES

## ABSTRACT

Efficient exploration is a key challenge in contextual bandits due to the potentially large size of their action space, where uninformed exploration can result in computational and statistical inefficiencies. Fortunately, the rewards of actions are often correlated and this can be leveraged to explore them efficiently. In this work, we capture such correlations using pre-trained linear diffusion models; upon which we design diffusion Thompson sampling (`dTS`). Both theoretical and algorithmic foundations are developed for `dTS`, and empirical evaluation also shows its favorable performance.

## 1 INTRODUCTION

A *contextual bandit* is a popular and practical framework for online learning under uncertainty (Li et al., 2010). In each round, an agent observes a *context*, takes an *action*, and receives a *reward* based on the context and action. The goal is to maximize the expected cumulative reward over $n$ rounds, striking a balance between exploiting actions with high estimated rewards from available data and exploring other actions to improve current estimates. This trade-off is often addressed using either *upper confidence bound (UCB)* (Auer et al., 2002) or *Thompson sampling (TS)* (Scott, 2010).

The action space in contextual bandits is often large, resulting in less-than-optimal performance with standard exploration strategies. Luckily, actions often exhibit correlations, making efficient exploration possible as one action may inform the agent about other actions. Notably, Thompson sampling offers remarkable flexibility, allowing its integration with informative prior distributions (Hong et al., 2022b) that can capture these correlations. Inspired by the achievements of diffusion models (Sohl-Dickstein et al., 2015; Ho et al., 2020), which effectively approximate complex distributions and enjoy state-of-the-art data generation performance (Dhariwal & Nichol, 2021; Rombach et al., 2022). This work focuses on capturing action correlations by employing linear diffusion models as priors in contextual Thompson sampling.

The idea is simple and illustrated through a video streaming scenario. The objective is to optimize watch time for a user $j$ by selecting a video $i \in [K]$, where $K$ is the number of videos. Users $j$ and videos $i$ are associated with context vectors $x_j$ and unknown video parameters $\theta_i$, respectively. User $j$'s expected watch time for video $i$ is linear as $x_j^\top \theta_i$. Then, a natural strategy would be independently learning video parameters $\theta_i$ using `LinTS` or `LinUCB` (Agrawal & Goyal, 2013a; Abbasi-Yadkori et al., 2011), but this proves statistically inefficient for larger $K$. Luckily, videos exhibit correlations and can provide informative insights into one another. To capture this, we leverage offline estimates of video parameters denoted by $\hat{\theta}_i$ and build a linear diffusion model on them. This linear diffusion model approximates the video parameter distribution, capturing their dependencies. This model enriches Thompson sampling as a prior, effectively capturing complex video dependencies while ensuring computational efficiency and theoretical guarantees thanks to closed-form posteriors.

Formally, we present a unified contextual bandit framework represented by diffusion models. On this basis, we design a computationally and statistically efficient Thompson sampling algorithm, called `dTS`. We then specialize `dTS` on linear instances, for which we provide closed-form solutions and establish an upper bound for its Bayes regret. The regret bound reflects the structure of the problem and the quality of the priors, demonstrating the benefits of using diffusion models as priors (`dTS`) over the standard methods such as linear Thompson sampling (`LinTS` Agrawal & Goyal (2013a)), linear UCB (`LinUCB` Abbasi-Yadkori et al. (2011)), etc. We also discuss the impact of the depth of

linear diffusion models, outlining their benefits through a sparsity concept. Note that, to broaden the applicability of our approach, dTS was provided for the general case, including non-linear diffusion models. However, approximate sampling techniques would be required for the latter, which we leave for future work, as we focus on linear diffusion models in this theoretical work. Finally, our empirical evaluations validate our theory and demonstrate the strong performance of dTS.

Diffusion models have been used for offline decision-making (Ajay et al., 2022; Janner et al., 2022; Wang et al., 2022). However, their use in online learning was only recently explored by Hsieh et al. (2023), who focused on *multi-armed bandits without theoretical guarantees*. Our work extends Hsieh et al. (2023) in two ways. First, we extend the idea to the broader contextual bandit framework. This allows us to consider problems where the rewards depend on the context, which is more realistic. Second, we show that when the diffusion model is parametrized by linear functions, we can derive recursive closed-form posteriors without the need for approximate sampling. Closed-form posteriors are particularly interesting because they facilitate theoretical analysis and improve computational efficiency; an important practical consideration, as it makes our algorithm more scalable. Finally, we provide a theoretical analysis of dTS, which effectively captures the benefits of employing linear diffusion models as priors within contextual Thompson sampling. This provides a complementary perspective to Hsieh et al. (2023), which focused on empirical evaluation rather than theoretical guarantees. An extended comparison to related works is provided in Appendix A, where we position our theoretical work with respect to the broader topics of diffusion models and decision-making, hierarchical, structured and low-rank bandits, approximate Thompson sampling, etc.

## 2 SETTING

The agent interacts with a *contextual bandit* environment over $n$ rounds. In round $t \in [n] = \{1, 2, ..., n\}$, the agent observes a *context* $X_t \in \mathcal{X}$, where $\mathcal{X} \subseteq \mathbb{R}^d$ is a $d$-dimensional *context space*, then it takes an *action* $A_t \in [K]$. Finally, the agent receives a stochastic reward $Y_t \in \mathbb{R}$ that depends on both the context $X_t$ and the taken action $A_t$. Each action $i \in [K]$ is associated with an *unknown action parameter* $\theta_{*,i} \in \mathbb{R}^d$, so that the reward received in round $t$ is $Y_t \sim P(\cdot \mid X_t; \theta_{*,A_t})$, where $P(\cdot \mid x; \theta_{*,a})$ is the reward distribution of action $a$ in context $x$. We consider the *Bayesian* bandit setting (Russo & Van Roy, 2014; Hong et al., 2022b), where the action parameters $\theta_{*,i}$ are assumed to be sampled from a known prior distribution, which we proceed to define using a diffusion model.

The correlations between the action parameters $\theta_{*,i}$ are captured through a diffusion model. Precisely, actions are correlated because they share a set of $L$ consecutive *unknown latent parameters* $\psi_{*,\ell}$ for $\ell \in [L]$. That is, the action parameter $\theta_{*,i}$ depends on the $L$-th latent parameter $\psi_{*,L}$ as $\theta_{*,i} \mid \psi_{*,1} \sim \mathcal{N}(f_1(\psi_{*,1}), \Sigma_1)$, where the mapping $f_1 : \mathbb{R}^d \to \mathbb{R}^d$ is *known*. Also, the $\ell - 1$-th latent parameter $\psi_{*,\ell-1}$ depends on the $\ell$-th latent parameter $\psi_{*,\ell}$ as $\psi_{*,\ell-1} \mid \psi_{*,\ell} \sim \mathcal{N}(f_\ell(\psi_{*,\ell}), \Sigma_\ell)$, where the mapping $f_\ell : \mathbb{R}^d \to \mathbb{R}^d$ is known. Finally, the $L$-th latent parameter $\psi_{*,L}$ is sampled as $\psi_{*,L} \sim \mathcal{N}(0, \Sigma_{L+1})$. The full model is described below, and its graphical representation is provided in Fig. 1.

$$\psi_{*,L} \sim \mathcal{N}(0, \Sigma_{L+1}), \qquad (1)$$
$$\psi_{*,\ell-1} \mid \psi_{*,\ell} \sim \mathcal{N}(f_\ell(\psi_{*,\ell}), \Sigma_\ell), \quad \forall \ell \in [L]/\{1\},$$
$$\theta_{*,i} \mid \psi_{*,1} \sim \mathcal{N}(f_1(\psi_{*,1}), \Sigma_1), \qquad \forall i \in [K],$$
$$Y_t \mid X_t, \theta_{*,A_t} \sim P(\cdot \mid X_t; \theta_{*,A_t}), \qquad \forall t \in [n].$$

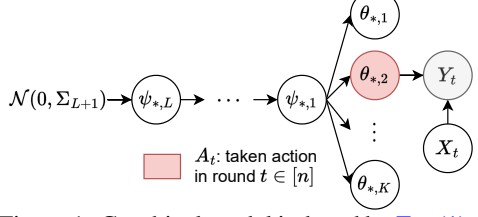

The model in Eq. (1) represents a Bayesian bandit. The agent interacts with a bandit instance defined by the action parameters $\theta_{*,i}$ over $n$ rounds (4-th line in Eq. (1)). These action parameters $\theta_{*,i}$ are drawn

Figure 1: Graphical model induced by Eq. (1).

from the generative process in the first 3 lines of Eq. (1). Note that Eq. (1) be built by pre-training a diffusion model on existing offline estimates of the action parameters $\theta_{*,i}$ (Hsieh et al., 2023).

A natural goal for the agent in a Bayesian framework is to minimize its *Bayes regret* (Russo & Van Roy, 2014) that measures the expected performance across multiple bandit instances $\theta_* = (\theta_{*,i})_{i \in [K]}$,

$$\mathcal{BR}(n) = \mathbb{E}\left[\sum_{t=1}^n r(X_t, A_{t,*}; \theta_*) - r(X_t, A_t; \theta_*)\right], \qquad (2)$$

where the expectation in Eq. (2) is taken over all quantities in Eq. (1) as well as the taken actions $A_t$. Here $r(x, i; \theta_*) = \mathbb{E}_{Y \sim P(\cdot|x;\theta_{*,i})}[Y]$ is the expected reward of action $i \in [K]$ in context $x \in \mathcal{X}$

and $A_{t,*} = \arg\max_{i \in [K]} r(X_t, i; \theta_*)$ is the optimal action in round $t$. The Bayes regret is known to capture the benefits of using informative priors, and hence it is suitable for our problem.

## 3 ALGORITHM

We design Thompson sampling that samples the latent and action parameters hierarchically (Lindley & Smith, 1972). Precisely, let $H_t = (X_k, A_k, Y_k)_{k \in [t-1]}$ be the history of all interactions up to round $t$ and let $H_{t,i} = (X_k, A_k, Y_k)_{\{k \in [t-1]; A_k = i\}}$ be the history of interactions *with action $i$* up to round $t$. To motivate our algorithm, we first decompose the posterior $\mathbb{P}(\theta_{*,i} = \theta \mid H_t)$ recursively as

$$\mathbb{P}(\theta_{*,i} = \theta \mid H_t) = \int_{\psi_L} \cdots \int_{\psi_1} Q_{t,L}(\psi_L) \prod_{\ell=2}^{L} Q_{t,\ell-1}(\psi_{\ell-1} \mid \psi_\ell) P_{t,i}(\theta \mid \psi_1) \, d\psi_1 \ldots d\psi_L, \quad (3)$$

where $Q_{t,L}(\psi_L) = \mathbb{P}(\psi_{*,L} = \psi_L \mid H_t)$ is the *latent-posterior* density of $\psi_{*,L} \mid H_t$. Moreover, for any $\ell \in [L]/\{1\}$, $Q_{t,\ell-1}(\psi_{\ell-1} \mid \psi_\ell) = \mathbb{P}(\psi_{*,\ell-1} = \psi_{\ell-1} \mid H_t, \psi_{*,\ell} = \psi_\ell)$ is the *conditional latent-posterior* density of $\psi_{*,\ell-1} \mid H_t, \psi_{*,\ell} = \psi_\ell$. Finally, for any action $i \in [K]$, $P_{t,i}(\theta \mid \psi_1) = \mathbb{P}(\theta_{*,i} = \theta \mid H_{t,i}, \psi_{*,1} = \psi_1)$ is the *conditional action-posterior* density of $\theta_{*,i} \mid H_{t,i}, \psi_{*,1} = \psi_1$.

The decomposition in Eq. (3) inspires hierarchical sampling. In each round $t \in [n]$, we initially sample the $L$-th latent parameter as $\psi_{t,L} \sim Q_{t,L}(\cdot)$. Then, for any $\ell \in [L]/\{1\}$, we sample the $\ell - 1$-th latent parameter given that $\psi_{*,\ell} = \psi_{t,\ell}$, as $\psi_{t,\ell-1} \sim Q_{t,\ell-1}(\cdot \mid \psi_{t,\ell})$. Lastly, given that $\psi_{*,1} = \psi_{t,1}$, each action parameter is sampled *individually* as $\theta_{t,i} \sim P_{t,i}(\theta \mid \psi_{t,1})$. This individual sampling is possible because action parameters $\theta_{*,i}$ are independent given $\psi_{*,1}$. This leads to Algorithm 1, named **d**iffusion **T**hompson **S**ampling (dTS). Our algorithm, dTS, requires sampling from the $K + L$ posteriors $P_{t,i}$ and $Q_{t,\ell}$. Thus we start by providing an efficient recursive scheme to *express* these posteriors for general diffusion models (Eq. (1)) using known quantities. We note that these expressions do not necessarily lead to closed-form posteriors and approximation might be needed. First, the conditional action-posterior $P_{t,i}(\cdot \mid \psi_1)$ writes

$$P_{t,i}(\theta \mid \psi_1) \propto \prod_{k \in S_{t,i}} P(Y_k \mid X_k; \theta) \mathcal{N}(\theta; f_1(\psi_1), \Sigma_1),$$

where $S_{t,i} = \{\ell \in [t-1] : A_\ell = i\}$ is the set of rounds where the agent takes action $i$ up to round $t$. Also, for any $\ell \in [L]/\{1\}$, the $\ell - 1$-th conditional latent-posterior $Q_{t,\ell-1}(\cdot \mid \psi_\ell)$ writes

$$Q_{t,\ell-1}(\psi_{\ell-1} \mid \psi_\ell) \propto \mathbb{P}(H_t \mid \psi_{*,\ell-1} = \psi_{\ell-1}) \mathcal{N}(\psi_{\ell-1}, f_\ell(\psi_\ell), \Sigma_\ell),$$

and the $L$-th latent-posterior is $Q_{t,L}(\cdot) = \mathcal{N}(\bar{\mu}_{t,L}, \bar{\Sigma}_{t,L}) \propto \mathbb{P}(H_t \mid \psi_{*,L} = \psi_L) \mathcal{N}(\psi_L, 0, \Sigma_{L+1})$. All the terms above are known, except $\mathbb{P}(H_t \mid \psi_{*,\ell} = \psi_\ell)$ for $\ell \in [L]$, which are the likelihoods of all observations up to round $t$ given that $\psi_{*,\ell} = \psi_\ell$. These likelihoods are computed recursively as follows. First, the basis of the recursion writes

$$\mathbb{P}(H_t \mid \psi_{*,1} = \psi_1) = \prod_{i \in [K]} \int_{\theta_i} \prod_{k \in S_{t,i}} P(Y_k \mid X_k; \theta_i) \mathcal{N}(\theta_i; f_1(\psi_1), \Sigma_1) \, d\theta_i.$$

Then for any $\ell \in [L]/\{1\}$, the recursive step follows as

$$\mathbb{P}(H_t \mid \psi_{*,\ell} = \psi_\ell) = \int_{\psi_{\ell-1}} \mathbb{P}(H_t \mid \psi_{*,\ell-1} = \psi_{\ell-1}) \mathcal{N}(\psi_{\ell-1}; f_\ell(\psi_\ell), \Sigma_\ell) \, d\psi_{\ell-1}.$$

All posterior expressions above use known quantities ($f_\ell$, $\Sigma_\ell$ and $P(y \mid x; \theta)$). However, these expressions either lead to closed-form solutions or should be approximated, depending on the form of functions $f_\ell$ and reward distribution $P(\cdot \mid x; \theta)$. In this work, we focus on theoretical guarantees and computational efficiency. Thus, we specialize dTS on cases where the functions $f_\ell$ are linear. In Section 3.1, we find closed-form posteriors for linear diffusion models with linear rewards. Then, in Section 3.2, we offer an approximation for linear diffusion models with non-linear rewards.

### 3.1 LINEAR DIFFUSION MODELS WITH LINEAR REWARDS

Here we we suppose that the functions $f_\ell$ are linear such as $f_\ell(\psi_{*,\ell}) = W_\ell \psi_{*,\ell}$ for any $\ell \in [L]$, where $W_\ell \in \mathbb{R}^{d \times d}$ are the *mixing matrices* and they are known. We also assume that the reward distribution is linear-Gaussian and writes $P(\cdot \mid x; \theta_{*,a}) = \mathcal{N}(\cdot; x^\top \theta_{*,a}, \sigma^2)$ where $\sigma > 0$ is the observation noise. Then, the model in Eq. (1) becomes

$$\psi_{*,L} \sim \mathcal{N}(0, \Sigma_{L+1}), \quad\quad\quad\quad\quad\quad\quad\quad (4)$$
$$\psi_{*,\ell-1} \mid \psi_{*,\ell} \sim \mathcal{N}(W_\ell \psi_{*,\ell}, \Sigma_\ell), \quad\quad\quad\quad \forall \ell \in [L]/\{1\},$$
$$\theta_{*,i} \mid \psi_{*,1} \sim \mathcal{N}(W_1 \psi_{*,1}, \Sigma_1), \quad\quad\quad\quad \forall i \in [K],$$
$$Y_t \mid X_t, \theta_{*,A_t} \sim \mathcal{N}(X_t^\top \theta_{*,A_t}, \sigma^2), \quad\quad\quad \forall t \in [n].$$

---

**Algorithm 1** dTS: **d**iffusion **T**hompson **S**ampling

---

**Input:** Prior information: $f_\ell$ for $\ell \in [L]$ and $\Sigma_\ell$ for $\ell \in [L+1]$.
**for** $t = 1, \ldots, n$ **do**
    Sample $\psi_{t,L} \sim Q_{t,L}$
    **for** $\ell = L, \ldots, 2$ **do**
        Sample $\psi_{t,\ell-1} \sim Q_{t,\ell-1}(\cdot \mid \psi_{t,\ell})$
    **for** $i = 1, \ldots, K$ **do**
        Sample $\theta_{t,i} \sim P_{t,i}(\cdot \mid \psi_{t,1})$
    Take action $A_t = \arg\max_{i \in [K]} X_t^\top \theta_{t,i}$, receive reward $Y_t$, update posteriors $Q_{t+1,\ell}$ and $P_{t+1,i}$.

---

The model in Eq. (4) is important, both for theory and practice, because it yields closed-form solutions. This leads to computationally efficient algorithms that are feasible for analysis. Now we derive the posteriors for this model and the proofs are provided in Appendix B. First, let $t \in [n]$, we introduce

$$\hat{G}_{t,i} = \sigma^{-2} \sum_{k \in S_{t,i}} X_k X_k^\top \in \mathbb{R}^{d \times d}, \qquad \hat{B}_{t,i} = \sigma^{-2} \sum_{k \in S_{t,i}} Y_k X_k \in \mathbb{R}^d,$$

where $S_{t,i} = \{\ell \in [t-1] : A_\ell = i\}$ is the set of rounds where the agent takes action $i$ up to round $t$. Then, the conditional action-posterior reads $P_{t,i}(\cdot \mid \psi_1) = \mathcal{N}(\cdot; \hat{\mu}_{t,i}, \hat{\Sigma}_{t,i})$, with

$$\hat{\Sigma}_{t,i}^{-1} = \Sigma_1^{-1} + \hat{G}_{t,i}, \qquad \hat{\mu}_{t,i} = \hat{\Sigma}_{t,i}(\Sigma_1^{-1} W_1 \psi_1 + \hat{B}_{t,i}). \tag{5}$$

For $\ell \in [L]/\{1\}$, the $\ell - 1$-th conditional latent-posterior is $Q_{t,\ell-1}(\cdot \mid \psi_\ell) = \mathcal{N}(\bar{\mu}_{t,\ell-1}, \bar{\Sigma}_{t,\ell-1})$,

$$\bar{\Sigma}_{t,\ell-1}^{-1} = \Sigma_\ell^{-1} + \bar{G}_{t,\ell-1}, \qquad \bar{\mu}_{t,\ell-1} = \bar{\Sigma}_{t,\ell-1}(\Sigma_\ell^{-1} W_\ell \psi_\ell + \bar{B}_{t,\ell-1}), \tag{6}$$

and the $L$-th latent-posterior reads $Q_{t,L}(\cdot) = \mathcal{N}(\bar{\mu}_{t,L}, \bar{\Sigma}_{t,L})$, with

$$\bar{\Sigma}_{t,L}^{-1} = \Sigma_{L+1}^{-1} + \bar{G}_{t,L}, \qquad \bar{\mu}_{t,L} = \bar{\Sigma}_{t,L} \bar{B}_{t,L}. \tag{7}$$

$\bar{G}_{t,\ell}$ and $\bar{B}_{t,\ell}$ for $\ell \in [L]$ are computed recursively. For $\ell = 1$, $\bar{G}_{t,1}$ and $\bar{B}_{t,1}$ are decomposed as

$$\bar{G}_{t,1} = W_1^\top \sum_{i \in [K]} (\Sigma_1^{-1} - \Sigma_1^{-1} \hat{\Sigma}_{t,i} \Sigma_1^{-1}) W_1, \qquad \bar{B}_{t,1} = W_1^\top \Sigma_1^{-1} \sum_{i \in [K]} \hat{\Sigma}_{t,i} \hat{B}_{t,i}. \tag{8}$$

Then, the recursive step follows from the fact that for $\ell \in [L]/\{1\}$,

$$\bar{G}_{t,\ell} = W_\ell^\top (\Sigma_\ell^{-1} - \Sigma_\ell^{-1} \bar{\Sigma}_{t,\ell-1} \Sigma_\ell^{-1}) W_\ell, \qquad \bar{B}_{t,\ell} = W_\ell^\top \Sigma_\ell^{-1} \bar{\Sigma}_{t,\ell-1} \bar{B}_{t,\ell-1}.$$

## 3.2 LINEAR DIFFUSION MODELS WITH NON-LINEAR REWARDS

Here all parameters $\psi_{*,\ell}$ and $\theta_{*,i}$ are generated using a linear diffusion model as in Eq. (4), except that the reward distribution is now parametrized as a generalized linear model (GLM) (McCullagh & Nelder, 1989). That is, for any $x \in \mathcal{X}$, $P(\cdot \mid x; \theta)$ is an exponential-family distribution with mean $g(x^\top \theta)$, where $g$ is the mean function. For example, let $g(u) = (1 + \exp(-u))^{-1}$ and $P(\cdot \mid x; \theta) = \text{Ber}(g(x^\top \theta))$, where $\text{Ber}(p)$ be the Bernoulli distribution with mean $p$. Then, this setting would correspond to a logistic bandit (Filippi et al., 2010). The full model writes

$$\begin{aligned}
\psi_{*,L} &\sim \mathcal{N}(0, \Sigma_{L+1}), & &(9) \\
\psi_{*,\ell-1} \mid \psi_{*,\ell} &\sim \mathcal{N}(W_\ell \psi_{*,\ell}, \Sigma_\ell), & \forall \ell \in [L]/\{1\}, \\
\theta_{*,i} \mid \psi_{*,1} &\sim \mathcal{N}(W_1 \psi_{*,1}, \Sigma_1), & \forall i \in [K], \\
Y_t \mid X_t, \theta_{*,A_t} &\sim P(\cdot \mid X_t; \theta_{*,A_t}), & \forall t \in [n].
\end{aligned}$$

Despite the linearity in latent parameters, we cannot derive closed-form posteriors here due to the non-linearity of the rewards. Therefore, we approximate the log-likelihoods $\log \mathbb{P}(H_{t,i} \mid \theta_{*,i} = \theta)$ by multivariate Gaussian densities using the Laplace approximation. Precisely, the reward function $P(\cdot \mid x; \theta)$ is an exponential-family distribution. Thus the log-likelihoods write $\log \mathbb{P}(H_{t,i} \mid \theta_{*,i} = \theta) = \sum_{k \in S_{t,i}} Y_k X_k^\top \theta - A(X_k^\top \theta) + C(Y_k)$, where $C$ is a real function, and $A$ is a twice continuously differentiable function whose derivative is the mean function, $\dot{A} = g$. Now let $\hat{\theta}_{t,i}^{\text{GLM}}$ and $\hat{G}_{t,i}^{\text{GLM}}$ be the maximum likelihood estimate (MLE) and the Hessian of the negative log-likelihood, respectively,

defined as $\hat{\theta}_{t,i}^{\text{GLM}} = \arg\max_{\theta \in \mathbb{R}^d} \log \mathbb{P}\left(H_{t,i} \mid \theta_{*,i} = \theta\right)$ and $\hat{G}_{t,i}^{\text{GLM}} = \sum_{k \in S_{t,i}} \dot{g}\left(X_k^\top \hat{\theta}_{t,i}^{\text{GLM}}\right) X_k X_k^\top$. The Laplace approximation follows as $\mathbb{P}\left(H_{t,i} \mid \theta_{*,i} = \theta\right) \approx \mathcal{N}\left(\theta; \hat{\theta}_{t,i}^{\text{GLM}}, \left(\hat{G}_{t,i}^{\text{GLM}}\right)^{-1}\right)$. Then we use the posteriors in Section 3.1, except that we replace $\hat{B}_{t,i}$ and $\hat{G}_{t,i}$ in Section 3.1 by $\hat{G}_{t,i}^{\text{GLM}} \hat{\theta}_{t,i}^{\text{GLM}}$ and $\hat{G}_{t,i}^{\text{GLM}}$, respectively. A question that may arise is why the Laplace approximation is applied to the likelihoods $\mathbb{P}\left(H_{t,i} \mid \theta_{*,i} = \theta\right)$ instead of the posteriors $\mathbb{P}\left(\theta_{*,i} = \theta \mid H_{t,i}\right)$, as is common in Bayesian inference. The reason behind this preference is that it allows us to use the same posterior derivations for linear rewards (Section 3.1) with slight adaptations, as explained before.

## 4 ANALYSIS

This section focuses on analyzing dTS under the linear diffusion model in Eq. (4). Although our result holds for milder assumptions, we make some simplifications for the sake of clarity and interpretability. We assume that **(A1)** Contexts satisfy $\|X_t\|_2^2 = 1$ for any $t \in [n]$. **(A2)** Mixing matrices and covariances satisfy $\lambda_1(W_\ell^\top W_\ell) = 1$ for any $\ell \in [L]$ and $\Sigma_\ell = \sigma_\ell^2 I_d$ for any $\ell \in [L+1]$. Note that **(A1)** can be relaxed to any contexts $X_t$ with bounded norms $\|X_t\|_2$. Also, **(A2)** can be relaxed to positive definite covariances $\Sigma_\ell$ and arbitrary mixing matrices $W_\ell$. In this section, we write $\tilde{\mathcal{O}}$ for the big-O notation up to polylogarithmic factors. Also, all proofs are provided in Appendix C. We start with the following standard lemma upon which we build our analysis (Aouali et al., 2023).

**Lemma 1.** *Assume that* $\mathbb{P}\left(\theta_{*,i} = \theta \mid H_t\right) = \mathcal{N}(\theta; \check{\mu}_{t,i}, \check{\Sigma}_{t,i})$ *for any* $i \in [K]$, *then for any* $\delta \in (0,1)$,

$$\mathcal{BR}(n) \leq \sqrt{2n \log(1/\delta)} \sqrt{\mathbb{E}\left[\sum_{t=1}^n \|X_t\|_{\check{\Sigma}_{t,A_t}}^2\right]} + cn\delta, \qquad \text{where } c > 0 \text{ is a constant}. \quad (10)$$

Applying Lemma 1 requires proving that the *marginal* action-posteriors $\mathbb{P}\left(\theta_{*,i} = \theta \mid H_t\right)$ in Eq. (3) are Gaussian and computing their covariances, while we only know the *conditional* action-posteriors $P_{t,i}$ and latent-posteriors $Q_{t,\ell}$. This is achieved by leveraging the preservation properties of the family of Gaussian distributions (Koller & Friedman, 2009) and the total covariance decomposition (Weiss, 2005) which leads to the next lemma.

**Lemma 2.** *Let* $t \in [n]$ *and* $i \in [K]$, *then the marginal covariance matrix* $\check{\Sigma}_{t,i}$ *reads*

$$\check{\Sigma}_{t,i} = \hat{\Sigma}_{t,i} + \sum_{\ell \in [L]} P_{i,\ell} \bar{\Sigma}_{t,\ell} P_{i,\ell}^\top, \quad \text{where } P_{i,\ell} = \hat{\Sigma}_{t,i} \Sigma_1^{-1} W_1 \prod_{k=1}^{\ell-1} \bar{\Sigma}_{t,k} \Sigma_{k+1}^{-1} W_{k+1}. \quad (11)$$

The marginal covariance matrix $\check{\Sigma}_{t,i}$ in Eq. (11) decomposes into $L+1$ terms. The first term corresponds to the posterior uncertainty of $\theta_{*,i} \mid \psi_{*,1}$. The remaining $L$ terms capture the posterior uncertainties of $\psi_{*,L}$ and $\psi_{*,\ell-1} \mid \psi_{*,\ell}$ for $\ell \in [L]/\{1\}$. These are then used to quantify the posterior information gain of latent parameters after one round as follows.

**Lemma 3** (Posterior information gain). *Let* $t \in [n]$ *and* $\ell \in [L]$, *then*

$$\bar{\Sigma}_{t+1,\ell}^{-1} - \bar{\Sigma}_{t,\ell}^{-1} \succeq \sigma^{-2} \sigma_{\text{MAX}}^{-2\ell} P_{A_t,\ell}^\top X_t X_t^\top P_{A_t,\ell}, \qquad \text{where } \sigma_{\text{MAX}}^2 = \max_{\ell \in [L+1]} 1 + \frac{\sigma_\ell^2}{\sigma^2}. \quad (12)$$

Finally, Lemma 2 is used to decompose $\|X_t\|_{\check{\Sigma}_{t,A_t}}^2$ in Eq. (10) into $L+1$ terms. Each term is bounded thanks to Lemma 3. This results in the following Bayes regret bound for dTS.

**Theorem 1.** *For any* $\delta \in (0,1)$, *the Bayes regret of* dTS *under Eq. (4),* **(A1)** *and* **(A2)** *is bounded as*

$$\mathcal{BR}(n) \leq \sqrt{2n\left(\mathcal{R}^{\text{ACT}}(n) + \sum_{\ell=1}^L \mathcal{R}_\ell^{\text{LAT}}\right) \log(1/\delta)} + cn\delta, \quad \text{where } c > 0 \text{ is a constant, and} \quad (13)$$

$$\mathcal{R}^{\text{ACT}}(n) = c_0 dK \log\left(1 + \frac{n\sigma_1^2}{d}\right), \ c_0 = \frac{\sigma_1^2}{\log(1+\sigma_1^2)}, \mathcal{R}_\ell^{\text{LAT}} = c_\ell d \log\left(1 + \frac{\sigma_{\ell+1}^2}{\sigma_\ell^2}\right), c_\ell = \frac{\sigma_{\ell+1}^2 \sigma_{\text{MAX}}^{2\ell}}{\log(1+\sigma_{\ell+1}^2)}.$$

Eq. (13) holds for any $\delta \in (0,1)$. In particular, the term $cn\delta$ is constant when $\delta = 1/n$. Then, the bound is $\tilde{\mathcal{O}}(\sqrt{n})$, and this dependence on the horizon $n$ aligns with prior Bayes regret bounds. The bound comprises $L+1$ main terms, $\mathcal{R}^{\text{ACT}}(n)$ and $\mathcal{R}_\ell^{\text{LAT}}$ for $\ell \in [L]$. First, $\mathcal{R}^{\text{ACT}}(n)$ relates to action parameters learning, conforming to a standard form (Lu & Van Roy, 2019). Similarly, $\mathcal{R}_\ell^{\text{LAT}}$ is associated with learning the $\ell$-th latent parameter. Roughly speaking, our bound captures that our problem can be seen as $L+1$ sequential linear bandit instances stacked upon each other.

To include more structure, we propose the *sparsity* assumption **(A3)** $W_\ell = (\bar{W}_\ell, 0_{d,d-d_\ell})$, where $\bar{W}_\ell \in \mathbb{R}^{d \times d_\ell}$ for any $\ell \in [L]$. Note that **(A3)** is not an assumption when $d_\ell = d$ for any $\ell \in [L]$. Notably, **(A3)** incorporates a plausible structural characteristic that could be captured by a diffusion model. Next we present Theorem 1 under **(A3)**.

**Proposition 1** (Sparsity). *For any $\delta \in (0,1)$, the Bayes regret of* dTS *under Eq. (4), **(A1)**, **(A2)** and **(A3)** is bounded as*

$$\mathcal{BR}(n) \le \sqrt{2n\big(\mathcal{R}^{\text{ACT}}(n) + \sum_{\ell=1}^{L} \tilde{\mathcal{R}}_\ell^{\text{LAT}}\big)\log(1/\delta)\big)} + cn\delta, \text{ where } c > 0 \text{ is a constant, and} \quad (14)$$

$$\mathcal{R}^{\text{ACT}}(n) = c_0 dK \log\big(1 + \tfrac{n\sigma_1^2}{d}\big), c_0 = \tfrac{\sigma_1^2}{\log(1+\sigma_1^2)}, \tilde{\mathcal{R}}_\ell^{\text{LAT}} = c_\ell d_\ell \log\big(1 + \tfrac{\sigma_{\ell+1}^2}{\sigma_\ell^2}\big), c_\ell = \tfrac{\sigma_{\ell+1}^2 \sigma_{\text{MAX}}^{2\ell}}{\log(1+\sigma_{\ell+1}^2)}.$$

From Proposition 1, the dependency of our bound with other parameters can be summarized as

$$\mathcal{BR}(n) = \tilde{\mathcal{O}}\Big(\sqrt{n(dK\sigma_1^2 + \sum_{\ell \in [L]} d_\ell \sigma_{\ell+1}^2 \sigma_{\text{MAX}}^{2\ell})}\Big), \quad (15)$$

since $\mathcal{R}^{\text{ACT}}(n) = \tilde{\mathcal{O}}(dK\sigma_1^2)$ and $\mathcal{R}_\ell^{\text{LAT}} = \tilde{\mathcal{O}}(d_\ell \sigma_{\ell+1}^2 \sigma_{\text{MAX}}^{2\ell})$. Then, smaller values of $K$, $L$, $d$ or $d_\ell$ translate to fewer parameters to learn, leading to lower regret. The regret also decreases when the initial variances $\sigma_\ell^2$ decrease. These dependencies are common in Bayesian analysis, and empirical results match them. They arise from the assumption that true parameters are sampled from a known distribution that matches our prior. When the prior is more informative (such as low variance), the problem is easier, resulting in lower Bayes regret. The reader might question the dependence of our bound on both $L$ and $K$, wondering why $K$ is present. This arises due to our conditional learning of $\theta_{*,i}$ given $\psi_{*,1}$. Rather than assuming deterministic linearity, $\theta_{*,i} = W_1 \psi_{*,1}$, we account for stochasticity by modeling $\theta_{*,i} \sim \mathcal{N}(W_1 \psi_{*,1}, \sigma_1^2 I_d)$. This makes dTS robust to misspecification scenarios where $\theta_{*,i}$ is not perfectly linear with respect to $\psi_{*,1}$, at the cost of additional learning of $\theta_{*,i} \mid \psi_{*,1}$. If we were to assume deterministic linearity ($\sigma_1 = 0$), our regret bound would scale with $L$ only and we provide this example in Section 4.1.

## 4.1 DISCUSSION AND COMPARISON TO STANDARD METHODS

Here we outline the statistical and computational merits of dTS, supported by theory and experiments.

**Computational benefits.** Action correlations prompts an intuitive approach: marginalize all latent parameters and maintain a joint posterior of $(\theta_{*,i})_{i \in [K]} \mid H_t$. Unfortunately, this is computationally inefficient for large action spaces. To illustrate, suppose that all posteriors are multivariate Gaussians (Section 3.1). Then maintaining the joint posterior $(\theta_{*,i})_{i \in [K]} \mid H_t$ necessitates converting and storing its $dK \times dK$-dimensional covariance matrix. Then the time and space complexities are $\mathcal{O}(K^3 d^3)$ and $\mathcal{O}(K^2 d^2)$. In contrast, the time and space complexities of dTS are $\mathcal{O}\big((L+K)d^3\big)$ and $\mathcal{O}\big((L+K)d^2\big)$. This is because dTS requires converting and storing $L + K$ covariance matrices, each being $d \times d$-dimensional. The improvement is huge when $K \gg L$, which is common in practice. Certainly, a more straightforward route to enhance computational efficiency is to discard latent parameters and maintain $K$ individual posteriors, each relating to an action parameter $\theta_{*,i} \in \mathbb{R}^d$ (LinTS). This improves time and space complexity to $\mathcal{O}(Kd^3)$ and $\mathcal{O}(Kd^2)$, correspondingly. However, LinTS maintaining independent posteriors fails to capture the correlations among actions; it only models $\theta_{*,i} \mid H_{t,i}$ rather than $\theta_{*,i} \mid H_t$ as done by dTS. Consequently, LinTS incurs higher regret due to the information loss caused by unused interactions of similar actions. Our regret bound and empirical results, which we will discuss next, reflect this aspect.

**Statistical benefits.** The linear diffusion model in Eq. (4) has a unique property. It can be transformed into a single Bayesian linear model (LinTS) by marginalizing out the latent parameters; in which case the prior on action parameters becomes $\theta_{*,i} \sim \mathcal{N}(0, \Sigma)$ for $i \in [K]$, with the $\theta_{*,i}$ being not necessarily independent, and $\Sigma$ is the marginal initial covariance of action parameters and it writes $\Sigma = \sigma_1^2 I_d + \sum_{\ell \in [L]} \sigma_{\ell+1}^2 B_\ell B_\ell^\top$ with $B_\ell = \prod_{k \in [\ell]} W_k$. Then, it is tempting to directly apply LinTS to solve our problem. While possible, this approach will suffer higher regret. To see this, note that the additional uncertainty of the latent parameters is accounted for in $\Sigma$ despite integrating them out. This causes the *marginal* action uncertainty $\Sigma$ to be much higher than the *conditional* action uncertainty $\sigma_1^2 I_d$ in Eq. (4), since we have $\Sigma = \sigma_1^2 I_d + \sum_{\ell \in [L]} \sigma_{\ell+1}^2 B_\ell B_\ell^\top \succcurlyeq \sigma_1^2 I_d$. This discrepancy leads to higher regret, particularly pronounced when $K$ is large. This is due to LinTS

needing to learn $K$ independent $d$-dimensional parameters, each with a considerably higher initial covariance $\Sigma$. This is also reflected by our regret bound. Precisely, the regret of LinTS scales as $\tilde{\mathcal{O}}\big(\sqrt{ndK(\sigma_1^2 + \sum_{\ell \in [L]} \sigma_{\ell+1}^2)}\big)$. This follows from Eq. (15) with $\sigma_\ell = 0$ for any $\ell \in [L+1]/\{1\}$, except that the maximum conditional action variance $\sigma_1^2$ is replaced with the maximum marginal action variance $\sigma_1^2 + \sum_{\ell \in [L]} \sigma_{\ell+1}^2$. The latter follows because $\sigma_1^2 I_d + \sum_{\ell \in [L]} \sigma_{\ell+1}^2 I_d \succcurlyeq \Sigma$. The same result can be obtained by applying the standard Bayes regret bound for LinTS. Now, let's compare the regret improvements of dTS compared to LinTS. To simplify, we assume that $\sigma \geq \max_{\ell \in [L+1]} \sigma_\ell$ so that $\sigma_{\text{MAX}}^2 \leq 2$. Then the regrets of dTS (where we bound $\sigma_{\text{MAX}}^{2\ell}$ by $2^\ell$) and LinTS are

$$\text{dTS} : \tilde{\mathcal{O}}\big(\sqrt{n(dK\sigma_1^2 + \sum_{\ell \in [L]} d_\ell \sigma_{\ell+1}^2 2^\ell)}\big), \quad \text{LinTS} : \tilde{\mathcal{O}}\big(\sqrt{ndK(\sigma_1^2 + \sum_{\ell \in [L]} \sigma_{\ell+1}^2)}\big).$$

Then regret improvements are captured by the variances $\sigma_\ell$ and the sparsity dimensions $d_\ell$, and we proceed to illustrate this through the following scenarios scenarios.

**(I) Decreasing variances.** Assume that $\sigma_\ell = 2^\ell$ for any $\ell \in [L+1]$. Then, the regrets become

$$\text{dTS} : \tilde{\mathcal{O}}\big(\sqrt{n(dK + \sum_{\ell \in [L]} d_\ell 4^\ell))}\big) \qquad \text{LinTS} : \tilde{\mathcal{O}}\big(\sqrt{ndK2^L}\big)$$

Now to see the order of gain, assume the problem is high-dimensional ($d \gg 1$), and set $L = \log_2(d)$ and $d_\ell = \lfloor \frac{d}{2^\ell} \rfloor$. Then the regret of dTS becomes $\tilde{\mathcal{O}}\big(\sqrt{nd(K + L)}\big)$, and hence the multiplicative factor $2^L$ in LinTS is removed and replaced with a smaller additive factor $L$.

**(II) Constant variances.** Assume that $\sigma_\ell = 1$ for any $\ell \in [L+1]$. Then, the regrets become

$$\text{dTS} : \tilde{\mathcal{O}}\big(\sqrt{n(dK + \sum_{\ell \in [L]} d_\ell 2^\ell))}\big) \qquad \text{LinTS} : \tilde{\mathcal{O}}\big(\sqrt{ndKL}\big)$$

Similarly, let $L = \log_2(d)$, and $d_\ell = \lfloor \frac{d}{2^\ell} \rfloor$. Then dTS's regret is $\tilde{\mathcal{O}}\big(\sqrt{nd(K + L)}\big)$. Thus the multiplicative factor $L$ in LinTS is removed and replaced with the additive factor $L$. By comparing this to **(I)**, the gain with decreasing variances is greater than with constant ones. In general, diffusion models use decreasing variances (Ho et al., 2020) and hence we expect great gains in practice.

**Effect of diffusion depth $L$.** The linear diffusion in Eq. (4) can be transformed into a two-level hierarchy (HierTS (Hong et al., 2022b)) using two different strategies. The first one yields,

$$\psi_{*,L} \sim \mathcal{N}(0, \sigma_{L+1}^2 B_L B_L^\top),$$
$$\text{HierTS-1}: \quad \theta_{*,i} \mid \psi_{*,L} \sim \mathcal{N}(\psi_{*,L}, \Omega_1), \qquad\qquad \forall i \in [K], \qquad\qquad (16)$$

where $\Omega_1 = \sigma_1^2 I_d + \sum_{\ell \in [L-1]} \sigma_{\ell+1}^2 B_\ell B_\ell^\top$ and $B_\ell = \prod_{k \in [\ell]} W_k$. The second one yields,

$$\psi_{*,1} \sim \mathcal{N}(0, \Omega_2),$$
$$\text{HierTS-2}: \quad \theta_{*,i} \mid \psi_{*,1} \sim \mathcal{N}(\psi_{*,1}, \sigma_1^2 I_d), \qquad\qquad \forall i \in [K], \qquad\qquad (17)$$

where $\Omega_2 = \sum_{\ell \in [L]} \sigma_{\ell+1}^2 B_\ell B_\ell^\top$. Then, we start by highlighting the differences between these two variants of HierTS. First, the regrets of HierTS-1 and HierTS-2 scale as

$$\text{HierTS-1} : \tilde{\mathcal{O}}\big(\sqrt{nd(K \sum_{\ell \in [L]} \sigma_\ell^2 + L\sigma_{L+1}^2}\big) \quad \text{HierTS-2} : \tilde{\mathcal{O}}\big(\sqrt{nd(K\sigma_1^2 + \sum_{\ell \in [L]} \sigma_{\ell+1}^2)}\big).$$

When $K$ and $L$ are comparable, then the regrets of HierTS-1 and HierTS-2 are similar. However, in more common scenarios where $K > L$, HierTS-2 tends to outperform HierTS-1. This superiority stems from HierTS-2's strategy of putting more uncertainty to the $d$-dimensional latent parameter $\psi_{*,1}$, rather than to $K$ individual $d$-dimensional action parameters $\theta_{*,i}$ for $i \in [K]$. Additionally, HierTS-1's regret is higher because it assumes that $\theta_{*,i}$ are conditionally independent given $\psi_{*,L}$, which may not always hold true. Consequently, HierTS-2 outperforms HierTS-1. Regarding dTS, and *in the absence of the sparsity assumption*, HierTS-2 and dTS essentially become equivalent. Specifically, their regrets' dependency on $K$ is identical, where both methods involve multiplying $K$ by $\sigma_1^2$, leading to improved performance compared to HierTS-1. In scenarios with dense mixing matrices, where assumption **(A3)** is not applicable, Theorem 1 indicates that dTS and HierTS-2 would demonstrate comparable levels of regret and computational efficiency. However, under the sparsity assumption and with certain mixing matrices that allow for conditional

Figure 2: Regret of two variants of `dTS`, `LindTS` (Section 3.1) and `GLM-dTS` (Section 3.2) on synthetic bandit problems with $d = 20$ and a varying number of actions $K \in \{10^3, 10^4\}$.

independence of $\psi_{*,1}$ coordinates given $\psi_{*,2}$, `dTS` acquires a great computational advantage over `HierTS-2`. This distinction explains why studies focusing on multi-level hierarchies typically benchmark their algorithms against two-level structures akin to `HierTS-1`, rather than the more competitive `HierTS-2`. This approach is consistent with previous works in Bayesian bandits using multi-level hierarchies, such as Tree-based priors (Hong et al., 2022a), which often favor comparisons with `HierTS-1`. In line with this, we also compared `dTS` with `HierTS-1` in our experiments.

Note that all observed improvements in this section could become even more pronounced when employing non-linear diffusion models. In our current analysis, we used linear diffusion models, and yet we can already discern substantial differences. Moreover, under non-linear diffusion models, the latent parameters cannot be analytically marginalized, making `LinTS` and `HierTS` inapplicable. We defer theoretical investigations of non-linear diffusion models to future works.

## 5 EXPERIMENTS

**Experimental setup.** We assess the performance of `dTS` using synthetic problems. Our primary objective is to validate our theory, and as a result, we do not include real-world datasets in our evaluation. This choice is further justified by the fact that Hsieh et al. (2023) has already demonstrated the advantages of diffusion models in multi-armed bandit settings using real-world datasets, without theoretical guarantees. We consider the two settings in Sections 3.1 and 3.2. The linear rewards (Section 3.1) are generated as $\mathcal{N}(x^\top \theta, \sigma^2)$ with $\sigma = 1$, and the non-linear rewards (Section 3.2) are binary and generated as $\mathrm{Ber}(g(x^\top \theta)))$, where $g$ is the sigmoid function. The covariances are $\Sigma_\ell = I_d$, and the context $X_t$ is uniformly drawn from $[-1, 1]^d$. The context dimension is $d = 20$ and the diffusion depth is $L = 4$. The mixing matrices $W_\ell$ are uniformly drawn from $[-1, 1]^{d \times d}$. To introduce sparsity, we zero out the last $d_\ell$ columns of $W_\ell$, resulting in $W_\ell = (\bar{W}_\ell, 0_{d, d-d_\ell})$, where $d_1 = 20$, $d_2 = 10$, $d_3 = 5$ and $d_4 = 2$ with $L = 4$. Also, we consider both $K = 10^3$ and $K = 10^4$. These values of $d$ and $K$ are high compared to experiments in prior works. We run 50 random simulations and plot the average regret alongside its standard error.

We consider several baselines. For linear rewards, we consider `dTS` under the linear diffusion model with linear rewards in Eq. (4). Second, `HierTS` (Hong et al., 2022b) that marginalizes out all latent parameters except $\psi_{*,L}$. Third, we include `LinUCB` (Abbasi-Yadkori et al., 2011) and `LinTS` (Agrawal & Goyal, 2013a). For non-linear rewards, we also use `dTS` under the linear diffusion model with non-linear rewards in Section 3.2, `UCB-GLM` (Li et al., 2017), and `GLM-TS` (Chapelle & Li, 2012). `GLM-UCB` (Filippi et al., 2010) induces a very high regret and thus it is not included.

**Results.** In Fig. 2, we present the regret values for various baseline methods over a horizon of $n = 5000$. We begin by examining the case of linear rewards, corresponding to the two plots on the left-hand side of Fig. 2. Our observations reveal that `LindTS` consistently outperforms all baselines that either disregard the latent structure (`LinTS` and `LinUCB`) or incorporate it only partially (`HierTS`). Specifically, the baselines that disregard the structure (`LinTS` and `LinUCB`) fail to converge in $n = 5000$ rounds since our problem involves a high-dimensional setting with $d = 20$ and a large action space of $K = 10^4$. These baselines appear to require a more extended horizon $n$ for convergence. Meanwhile, `HierTS`, which partially leverages the structure, manages to converge within the considered horizon but still incurs a significantly higher regret compared to `LindTS`. In contrast, `LindTS` demonstrates rapid convergence and maintains remarkably low

regret. Notably, we observe that as the action space grows (indicated by a higher value of $K$), the performance gap between `LindTS` and the other baselines widens.

In the case of non-linear rewards, illustrated in the two plots on the right-hand side of Fig. 2, `GLM-dTS` also surpasses all baselines. Moreover, `GLM-dTS` outperforms `LindTS` in this setting, underscoring the advantages of our approximation in Section 3.2. While `GLM-dTS` lacks theoretical guarantees, the results suggest that it is preferable to use `GLM-dTS` when dealing with non-linear rewards, emphasizing the generality and adaptability of `dTS` and the general posterior derivations in Section 3. However, despite the misspecification of the reward model in `LindTS`, it still outperforms models that use the correct reward model but neglect the latent structure, such as `GLM-TS` and `UCB-GLM`. This highlights the importance of accounting for the latent structure, which can outweigh the correctness of the reward model itself in some cases. This observation becomes particularly pronounced by noticing that all other baselines that assume linear rewards (`LinTS`, `LinUCB`, `HierTS`) induce high regret in this setting, with the exception of `LindTS` that correctly uses the latent structure. Finally, we also conduct an additional experiment to verify the relationships outlined in Theorem 1 between the regret of `dTS` and the number of actions $K$, the context dimension $d$, and the diffusion depth $L$. The results are provided in Appendix D and they match our theory.

## 6 CONCLUSION

In practice, grappling with large action spaces in contextual bandits is challenging. Recognizing this, we focused on structured contextual bandit problems where action parameters are sampled from a diffusion model; upon which we built diffusion Thompson sampling (`dTS`). Our primary contribution lies in proving Bayes regret bounds that quantify the statistical efficiency gains achieved by `dTS` when compared to conventional methods such as linear Thompson sampling `LinTS` (Agrawal & Goyal, 2013a), linear UCB (Abbasi-Yadkori et al., 2011), etc. This presents an important advancement in the theoretical comprehension of bandit algorithms capable of effectively navigating complex graphical model, such as linear diffusion models.

We identified several directions for future work. Notably, extending our work to non-linear diffusion models, both from a theoretical and empirical standpoint. From a theoretical perspective, future research could explore the advantages of non-linear diffusion models by deriving their Bayes regret bounds, akin to our analysis in Section 4. This would showcase the benefits of incorporating non-linearity into our framework. Empirically, investigating the use of approximate Thompson sampling techniques for non-linear diffusion models is interesting. Although we express the posteriors in the general case in Section 3, approximating these expressions is not straightforward. Additionally, note that while Hsieh et al. (2023) provided an approximation for Thompson sampling under non-linear diffusion priors in multi-armed bandits, extending this approximation to contextual bandits is challenging. Addressing these challenges and enabling the application of such approximations in contextual bandits is an important direction of future work.

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

SUPPLEMENTARY MATERIALS

**Notation.** For any positive integer $n$, we define $[n] = \{1, 2, ..., n\}$. Let $v_1, \ldots, v_n \in \mathbb{R}^d$ be $n$ vectors, $(v_i)_{i \in [n]} \in \mathbb{R}^{nd}$ is the $nd$-dimensional vector obtained by concatenating $v_1, \ldots, v_n$. For any matrix $A \in \mathbb{R}^{d \times d}$, $\lambda_1(A)$ and $\lambda_d(A)$ denote the maximum and minimum eigenvalues of A, respectively. Finally, we write $\tilde{\mathcal{O}}$ for the big-O notation up to polylogarithmic factors.

## A    EXTENDED RELATED WORK

**Thompson sampling (TS)** operates within the Bayesian framework and it involves specifying a prior/likelihood model. In each round, the agent samples unknown model parameters from the current posterior distribution. The chosen action is the one that maximizes the resulting reward. TS is naturally randomized, particularly simple to implement, and has highly competitive empirical performance in both simulated and real-world problems (Russo & Van Roy, 2014; Chapelle & Li, 2012). Regret guarantees for the TS heuristic remained open for decades even for simple models. Recently, however, significant progress has been made. For standard multi-armed bandits, TS is optimal in the Beta-Bernoulli model (Kaufmann et al., 2012; Agrawal & Goyal, 2013b), Gaussian-Gaussian model (Agrawal & Goyal, 2013b), and in the exponential family using Jeffrey's prior (Korda et al., 2013). For linear bandits, TS is nearly-optimal (Russo & Van Roy, 2014; Agrawal & Goyal, 2017; Abeille & Lazaric, 2017). In this work, we build TS upon complex diffusion priors and analyze the resulting Bayes regret (Russo & Van Roy, 2014) in the linear contextual bandit setting.

**Decision-making with diffusion models** gained attention recently, especially in offline learning (Ajay et al., 2022; Janner et al., 2022; Wang et al., 2022). However, their application in online learning was only examined by Hsieh et al. (2023), which focused on meta-learning in multi-armed bandits without theoretical guarantees. In this work, we expand the scope of Hsieh et al. (2023) to encompass the broader contextual bandit framework. In particular, we provide theoretical analysis for linear instances, effectively capturing the advantages of using diffusion models as priors in contextual Thompson sampling. These linear cases are particularly captivating due to closed-form posteriors, enabling both theoretical analysis and computational efficiency; an important practical consideration.

**Hierarchical Bayesian bandits** (Bastani et al., 2019; Kveton et al., 2021; Basu et al., 2021; Simchowitz et al., 2021; Wan et al., 2021; Hong et al., 2022b; Peleg et al., 2022; Wan et al., 2022; Aouali et al., 2023) applied TS to simple graphical models, wherein action parameters are generally sampled from a Gaussian distribution centered at a single latent parameter. These works mostly span meta- and multi-task learning for multi-armed bandits, except in cases such as Aouali et al. (2023); Hong et al. (2022a) that consider the contextual bandit setting. Precisely, Aouali et al. (2023) assume that action parameters are sampled from a Gaussian distribution centered at a linear mixture of multiple latent parameters. On the other hand, Hong et al. (2022a) applied TS to a graphical model represented by a tree. Our work can be seen as an extension of all these works to much more complex graphical models, for which both theoretical and algorithmic foundations are developed. Note that the settings in most of these works can be recovered with specific choices of the diffusion depth $L$ and functions $f_\ell$. This attests to the modeling power of dTS.

**Approximate Thompson sampling** is a major problem in the Bayesian inference literature. This is because most posterior distributions are intractable, and thus practitioners must resort to sophisticated computational techniques such as Markov chain Monte Carlo (Kruschke, 2010). Prior works (Riquelme et al., 2018; Chapelle & Li, 2012; Kveton et al., 2020) highlight the favorable empirical performance of approximate Thompson sampling. Particularly, (Kveton et al., 2020) provide theoretical guarantees for Thompson sampling when using the Laplace approximation in generalized linear bandits (GLB). In our context, we incorporate approximate sampling when the reward exhibits non-linearity. While our approximation does not come with formal guarantees, it enjoys strong practical performance. An in-depth analysis of this approximation is left as a direction for future works. Similarly, approximating the posterior distribution when the diffusion model is non-linear as well as analyzing it is an interesting direction of future works.

**Bandits with underlying structure** also align with our work, where we assume a structured relationship among actions, captured by a diffusion model. In latent bandits (Maillard & Mannor, 2014; Hong et al., 2020), a single latent variable indexes multiple candidate models. Within structured

finite-armed bandits (Lattimore & Munos, 2014; Gupta et al., 2018), each action is linked to a known mean function parameterized by a common latent parameter. This latent parameter is learned. TS was also applied to complex structures (Yu et al., 2020; Gopalan et al., 2014). However, simultaneous computational and statistical efficiencies aren't guaranteed. Meta- and multi-task learning with upper confidence bound (UCB) approaches have a long history in bandits (Azar et al., 2013; Gentile et al., 2014; Deshmukh et al., 2017; Cella et al., 2020). These, however, often adopt a frequentist perspective, analyze a stronger form of regret, and sometimes result in conservative algorithms. In contrast, our approach is Bayesian, with analysis centered on Bayes regret. Remarkably, our algorithm, dTS, performs well as analyzed without necessitating additional tuning. Finally, **Low-rank bandits** (Hu et al., 2021; Cella et al., 2022; Yang et al., 2020) also relate to our linear diffusion model when $L = 1$. Broadly, there exist two key distinctions between these prior works and the special case of our model (linear diffusion model with $L = 1$). First, they assume $\theta_{*,i} = W_1 \psi_{*,1}$, whereas we incorporate additional uncertainty in the covariance $\Sigma_1$ to account for possible misspecification as $\theta_{*,i} = \mathcal{N}(W_1 \psi_{*,1}, \Sigma_1)$. Consequently, these algorithms might suffer linear regret due to model misalignment. Second, we assume that the mixing matrix $W_1$ is available and pre-learned offline, whereas they learn it online. While this is more general, it leads to computationally expensive methods that are difficult to employ in a real-world online setting.

**Large action spaces.** Roughly speaking, the regret bound of dTS scales with $K\sigma_1^2$ rather than $K \sum_\ell \sigma_\ell^2$. This is particularly beneficial when $\sigma_1$ is small, a common scenario in diffusion models with decreasing variances. A notable case is when $\sigma_1 = 0$, where the regret becomes independent of $K$. Also, our analysis (Section 4.1) indicates that the gap in performance between dTS and LinTS becomes more pronounced when the number of action increases, highlighting dTS's suitability for large action spaces. Note that some prior works (Foster et al., 2020; Xu & Zeevi, 2020; Zhu et al., 2022) proposed bandit algorithms that do not scale with $K$. However, our setting differs significantly from theirs, explaining our inherent dependency on $K$ when $\sigma_1 > 0$. Precisely, they assume a reward function of $r(x, i) = \phi(x, i)^\top \theta_*$, with a shared $\theta_* \in \mathbb{R}^d$ across actions and a known mapping $\phi$. In contrast, we consider $r(x, i) = x^\top \theta_{*,i}$, requiring the learning of $K$ separate $d$-dimensional action parameters. In their setting, with the availability of $\phi$, the regret of dTS would similarly be independent of $K$. However, obtaining such a mapping $\phi$ can be challenging as it needs to encapsulate complex context-action dependencies. Notably, our setting reflects a common practical scenario, such as in recommendation systems where each product is often represented by its embedding. In summary, the dependency on $K$ is more related to our setting than the method itself, and dTS would scale with $d$ only in their setting. Note that dTS is both computationally and statistically efficient (Section 4.1). This becomes particularly notable in large action spaces. Our empirical results in Fig. 2, notably with $K = 10^4$, demonstrate that dTS significantly outperforms the baselines. More importantly, the performance gap between dTS and these baselines is larger when the number of actions ($K$) increases, highlighting the improved scalability of dTS to large action spaces.

## B  DERIVATION OF CLOSED-FORM POSTERIORS FOR LINEAR DIFFUSION MODELS

In this section, we derive the $K + L$ posteriors $P_{t,i}$ and $Q_{t,\ell}$, for which we provide the expressions in Section 3.1. In our proofs, $p(x) \propto f(x)$ means that the probability density $p$ satisfies $p(x) = \frac{f(x)}{Z}$ for any $x \in \mathbb{R}^d$, where $Z$ is a normalization constant. In particular, we extensively use that if $p(x) \propto \exp[-\frac{1}{2}x^\top \Lambda x + x^\top m]$, where $\Lambda$ is positive definite. Then $p$ is the multivariate Gaussian density with covariance $\Sigma = \Lambda^{-1}$ and mean $\mu = \Sigma m$. These are standard notations and techniques to manipulate Gaussian distributions (Koller & Friedman, 2009, Chapter 7).

### B.1  DERIVATION OF THE ACTION-POSTERIOR FOR LINEAR DIFFUSION MODELS

**Proposition 2.** *Consider the following model, which corresponds to the last two layers in Eq. (4)*

$$\theta_{*,i} \mid \psi_{*,1} \sim \mathcal{N}\left(W_1 \psi_{*,1}, \Sigma_1\right),$$
$$Y_t \mid X_t, \theta_{*,A_t} \sim \mathcal{N}\left(X_t^\top \theta_{*,A_t}, \sigma^2\right), \qquad \forall t \in [n].$$

*Then we have that for any $t \in [n]$ and $i \in [K]$, $P_{t,i}(\theta \mid \psi_1) = \mathbb{P}(\theta_{*,i} = \theta \mid \psi_{*,1} = \psi_1, H_{t,i}) = \mathcal{N}(\theta; \hat{\mu}_{t,i}, \hat{\Sigma}_{t,i})$, where*

$$\hat{\Sigma}_{t,i}^{-1} = \hat{G}_{t,i} + \Sigma_1^{-1}, \qquad\qquad \hat{\mu}_{t,i} = \hat{\Sigma}_{t,i}\left(\hat{B}_{t,i} + \Sigma_1^{-1}W_1\psi_1\right).$$

*Proof.* Let $v = \sigma^{-2}, \quad \Lambda_1 = \Sigma_1^{-1}$. Then the action-posterior decomposes as

$$
\begin{aligned}
P_{t,i}(\theta \mid \psi_1) &= \mathbb{P}(\theta_{*,i} = \theta \mid \psi_{*,1} = \psi_1, H_{t,i}), \\
&\propto \mathbb{P}(H_{t,i} \mid \psi_{*,1} = \psi_1, \theta_{*,i} = \theta)\,\mathbb{P}(\theta_{*,i} = \theta \mid \psi_{*,1} = \psi_1), \quad \text{(Bayes rule)} \\
&= \mathbb{P}(H_{t,i} \mid \theta_{*,i} = \theta)\,\mathbb{P}(\theta_{*,i} = \theta \mid \psi_{*,1} = \psi_1), \text{(given } \theta_{*,i}, H_{t,i} \text{ is independent of } \psi_{*,1}) \\
&= \prod_{k \in S_{t,i}} \mathcal{N}(Y_k; X_k^\top\theta, \sigma^2)\mathcal{N}(\theta; W_1\psi_1, \Sigma_1), \\
&= \exp\Big[-\frac{1}{2}\Big(v\sum_{k \in S_{t,i}}(Y_k^2 - 2Y_kX_k^\top\theta + (X_k^\top\theta)^2) + \theta^\top\Lambda_1\theta - 2\theta^\top\Lambda_1W_1\psi_1 \\
&\qquad\qquad + \left(W_1\psi_1\right)^\top\Lambda_1\left(W_1\psi_1\right)\Big)\Big], \\
&\propto \exp\Big[-\frac{1}{2}\Big(\theta^\top(v\sum_{k \in S_{t,i}}X_kX_k^\top + \Lambda_1)\theta - 2\theta^\top\Big(v\sum_{k \in S_{t,i}}X_kY_k + \Lambda_1W_1\psi_1\Big)\Big)\Big], \\
&\propto \mathcal{N}\big(\theta; \hat{\mu}_{t,i}, \hat{\Lambda}_{t,i}^{-1}\big),
\end{aligned}
$$

where $\hat{\Lambda}_{t,i} = v\sum_{k \in S_{t,i}} X_kX_k^\top + \Lambda_1$, and $\hat{\Lambda}_{t,i}\hat{\mu}_{t,i} = v\sum_{k \in S_{t,i}} X_kY_k + \Lambda_1W_1\psi_1$. Using that $\hat{B}_{t,i} = v\sum_{k \in S_{t,i}} X_kY_k$ and $\hat{G}_{t,i} = v\sum_{k \in S_{t,i}} X_kX_k^\top$ concludes the proof. $\qquad\square$

### B.2 Derivation of the recursive latent-posteriors for linear diffusion models

**Proposition 3.** *For any $\ell \in [L]/\{1\}$, the $\ell-1$-th conditional latent-posterior reads $Q_{t,\ell-1}(\cdot \mid \psi_\ell) = \mathcal{N}(\bar{\mu}_{t,\ell-1}, \bar{\Sigma}_{t,\ell-1})$, with*

$$\bar{\Sigma}_{t,\ell-1}^{-1} = \Sigma_\ell^{-1} + \bar{G}_{t,\ell-1}, \qquad \bar{\mu}_{t,\ell-1} = \bar{\Sigma}_{t,\ell-1}\big(\Sigma_\ell^{-1}W_\ell\psi_\ell + \bar{B}_{t,\ell-1}\big), \qquad (18)$$

*and the $L$-th latent-posterior reads $Q_{t,L}(\cdot) = \mathcal{N}(\bar{\mu}_{t,L}, \bar{\Sigma}_{t,L})$, with*

$$\bar{\Sigma}_{t,L}^{-1} = \Sigma_{L+1}^{-1} + \bar{G}_{t,L}, \qquad\qquad \bar{\mu}_{t,L} = \bar{\Sigma}_{t,L}\bar{B}_{t,L}. \qquad (19)$$

*Proof.* Let $\ell \in [L]/\{1\}$. Then, Bayes rule yields that

$$Q_{t,\ell-1}(\psi_{\ell-1} \mid \psi_\ell) \propto \mathbb{P}(H_t \mid \psi_{*,\ell-1} = \psi_{\ell-1})\,\mathcal{N}(\psi_{\ell-1}, W_\ell\psi_\ell, \Sigma_\ell),$$

But from [Lemma 4](), we know that

$$\mathbb{P}(H_t \mid \psi_{*,\ell-1} = \psi_{\ell-1}) \propto \exp\Big[-\frac{1}{2}\psi_{\ell-1}^\top\bar{G}_{t,\ell-1}\psi_{\ell-1} + \psi_{\ell-1}^\top\bar{B}_{t,\ell-1}\Big].$$

Therefore,

$$
\begin{aligned}
Q_{t,\ell-1}(\psi_{\ell-1} \mid \psi_\ell) &\propto \exp\Big[-\frac{1}{2}\psi_{\ell-1}^\top\bar{G}_{t,\ell-1}\psi_{\ell-1} + \psi_{\ell-1}^\top\bar{B}_{t,\ell-1}\Big]\mathcal{N}(\psi_{\ell-1}, W_\ell\psi_\ell, \Sigma_\ell), \\
&\propto \exp\Big[-\frac{1}{2}\psi_{\ell-1}^\top\bar{G}_{t,\ell-1}\psi_{\ell-1} + \psi_{\ell-1}^\top\bar{B}_{t,\ell-1} \\
&\qquad\qquad -\frac{1}{2}(\psi_{\ell-1} - W_\ell\psi_\ell)^\top\Sigma_\ell^{-1}(\psi_{\ell-1} - W_\ell\psi_\ell))\Big], \\
&\overset{(i)}{\propto} \exp\Big[-\frac{1}{2}\psi_{\ell-1}^\top(\bar{G}_{t,\ell-1} + \Sigma_\ell^{-1})\psi_{\ell-1} + \psi_{\ell-1}^\top(\bar{B}_{t,\ell-1} + \Sigma_\ell^{-1}W_\ell\psi_\ell)\Big], \\
&\overset{(ii)}{\propto} \mathcal{N}(\psi_{\ell-1}; \bar{\mu}_{t,\ell-1}, \bar{\Sigma}_{t,\ell-1}),
\end{aligned}
$$

with $\bar{\Sigma}_{t,\ell-1}^{-1} = \Sigma_\ell^{-1} + \bar{G}_{t,\ell-1}$ and $\bar{\mu}_{t,\ell-1} = \bar{\Sigma}_{t,\ell-1}\left(\Sigma_\ell^{-1}W_\ell\psi_\ell + \bar{B}_{t,\ell-1}\right)$. In $(i)$, we omit terms that are constant in $\psi_{\ell-1}$. In $(ii)$, we complete the square. This concludes the proof for $\ell \in [L]/\{1\}$. For $Q_{t,L}$, we use Bayes rule to get

$$Q_{t,L}(\psi_L) \propto \mathbb{P}\left(H_t \mid \psi_{*,L} = \psi_L\right) \mathcal{N}(\psi_L, 0, \Sigma_{L+1}).$$

Then from Lemma 4, we know that

$$\mathbb{P}\left(H_t \mid \psi_{*,L} = \psi_L\right) \propto \exp\left[-\frac{1}{2}\psi_L^\top \bar{G}_{t,L}\psi_L + \psi_L^\top \bar{B}_{t,L}\right],$$

We then use the same derivations above to compute the product $\exp\left[-\frac{1}{2}\psi_L^\top \bar{G}_{t,L}\psi_L + \psi_L^\top \bar{B}_{t,L}\right] \times \mathcal{N}(\psi_L, 0, \Sigma_{L+1})$, which concludes the proof. $\qquad\square$

**Lemma 4.** *The following holds for any $t \in [n]$ and $\ell \in [L]$,*

$$\mathbb{P}\left(H_t \mid \psi_{*,\ell} = \psi_\ell\right) \propto \exp\left[-\frac{1}{2}\psi_\ell^\top \bar{G}_{t,\ell}\psi_\ell + \psi_\ell^\top \bar{B}_{t,\ell}\right],$$

*where $\bar{G}_{t,\ell}$ and $\bar{B}_{t,\ell}$ are defined by recursion in Section 3.1.*

*Proof.* We prove this result by induction. To reduce clutter, we let $v = \sigma^{-2}$, and $\Lambda_1 = \Sigma_1^{-1}$. We start with the base case of the induction when $\ell = 1$.

**(I) Base case.** Here we want to show that $\mathbb{P}\left(H_t \mid \psi_{*,1} = \psi_1\right) \propto \exp\left[-\frac{1}{2}\psi_1^\top \bar{G}_{t,1}\psi_1 + \psi_1^\top \bar{B}_{t,1}\right]$, where $\bar{G}_{t,1}$ and $\bar{B}_{t,1}$ are given in Eq. (8). First, we have that

$$\mathbb{P}\left(H_t \mid \psi_{*,1} = \psi_1\right) \overset{(i)}{=} \prod_{i\in[K]} \mathbb{P}\left(H_{t,i} \mid \psi_{*,1} = \psi_1\right) = \prod_{i\in[K]} \int_\theta \mathbb{P}\left(H_{t,i}, \theta_{*,i} = \theta \mid \psi_{*,1} = \psi_1\right) \mathrm{d}\theta,$$

$$= \prod_{i\in[K]} \int_\theta \mathbb{P}\left(H_{t,i} \mid \theta_{*,i} = \theta\right) \mathcal{N}\left(\theta; W_1\psi_1, \Sigma_1\right) \mathrm{d}\theta,$$

$$= \prod_{i\in[K]} \underbrace{\int_\theta \left(\prod_{k\in S_{t,i}} \mathcal{N}(Y_k; X_k^\top\theta, \sigma^2)\right) \mathcal{N}\left(\theta; W_1\psi_1, \Sigma_1\right) \mathrm{d}\theta}_{h_i(\psi_1)},$$

$$= \prod_{i\in[K]} h_i(\psi_1), \tag{20}$$

where $(i)$ follows from the fact that $\theta_{*,i}$ for $i \in [K]$ are conditionally independent given $\psi_{*,1} = \psi_1$ and that given $\theta_{*,i}$, $H_{t,i}$ is independent of $\psi_{*,1}$. Now we compute $h_i(\psi_1) = \int_\theta \left(\prod_{k\in S_{t,i}} \mathcal{N}(Y_k; X_k^\top\theta, \sigma^2)\right) \mathcal{N}\left(\theta; W_1\psi_1, \Sigma_1\right) \mathrm{d}\theta$ as

$$h_i(\psi_1) = \int_\theta \left(\prod_{k\in S_{t,i}} \mathcal{N}(Y_k; X_k^\top\theta, \sigma^2)\right) \mathcal{N}(\theta; W_1\psi_1, \Sigma_1) \mathrm{d}\theta,$$

$$\propto \int_\theta \exp\left[-\frac{1}{2}v\sum_{k\in S_{t,i}}(Y_k - X_k^\top\theta)^2 - \frac{1}{2}(\theta - W_1\psi_1)^\top\Lambda_1(\theta - W_1\psi_1)\right]\mathrm{d}\theta,$$

$$= \int_\theta \exp\left[-\frac{1}{2}\left(v\sum_{k\in S_{t,i}}(Y_k^2 - 2Y_k\theta^\top X_k + (\theta^\top X_k)^2) + \theta^\top\Lambda_1\theta - 2\theta^\top\Lambda_1 W_1\psi_1\right.\right.$$

$$\left.\left. + (W_1\psi_1)^\top\Lambda_1(W_1\psi_1)\right)\right]\mathrm{d}\theta,$$

$$\propto \int_\theta \exp\left[-\frac{1}{2}\left(\theta^\top\left(v\sum_{k\in S_{t,i}}X_kX_k^\top + \Lambda_1\right)\theta - 2\theta^\top\left(v\sum_{k\in S_{t,i}}Y_kX_k\right.\right.\right.$$

$$\left.\left.\left. + \Lambda_1 W_1\psi_1\right) + (W_1\psi_1)^\top\Lambda_1(W_1\psi_1)\right)\right]\mathrm{d}\theta.$$

But we know that $\hat{G}_{t,i} = v \sum_{k \in S_{t,i}} X_k X_k^\top$, and $\hat{B}_{t,i} = v \sum_{k \in S_{t,i}} Y_k X_k$. To further simplify expressions, we also let

$$V = \left(\hat{G}_{t,i} + \Lambda_1\right)^{-1}, \quad U = V^{-1}, \quad \beta = V\left(\hat{B}_{t,i} + \Lambda_1 W_1 \psi_1\right).$$

We have that $UV = VU = I_d$, and thus

$$h_i(\psi_1) \propto \int_\theta \exp\left[-\frac{1}{2}\left(\theta^\top U \theta - 2\theta^\top UV\left(\hat{B}_{t,i} + \Lambda_1 W_1 \psi_1\right) + (W_1\psi_1)^\top \Lambda_1(W_1\psi_1)\right)\right] d\theta,$$

$$= \int_\theta \exp\left[-\frac{1}{2}\left(\theta^\top U \theta - 2\theta^\top U\beta + (W_1\psi_1)^\top \Lambda_1(W_1\psi_1)\right)\right] d\theta,$$

$$= \int_\theta \exp\left[-\frac{1}{2}\left((\theta - \beta)^\top U(\theta - \beta) - \beta^\top U\beta + (W_1\psi_1)^\top \Lambda_1(W_1\psi_1)\right)\right] d\theta,$$

$$\propto \exp\left[-\frac{1}{2}\left(-\beta^\top U\beta + (W_1\psi_1)^\top \Lambda_1(W_1\psi_1)\right)\right],$$

$$= \exp\left[-\frac{1}{2}\left(-\left(\hat{B}_{t,i} + \Lambda_1 W_1 \psi_1\right)^\top V\left(\hat{B}_{t,i} + \Lambda_1 W_1 \psi_1\right) + (W_1\psi_1)^\top \Lambda_1(W_1\psi_1)\right)\right],$$

$$\propto \exp\left[-\frac{1}{2}\left(\psi_1^\top W_1^\top (\Lambda_1 - \Lambda_1 V \Lambda_1) W_1 \psi_1 - 2\psi_1^\top \left(W_1^\top \Lambda_1 V \hat{B}_{t,i}\right)\right)\right],$$

$$= \exp\left[-\frac{1}{2}\psi_1^\top \Omega_i \psi_1 + \psi_1^\top m_i\right],$$

where

$$\Omega_i = W_1^\top (\Lambda_1 - \Lambda_1 V \Lambda_1) W_1 = W_1^\top \left(\Lambda_1 - \Lambda_1(\hat{G}_{t,i} + \Lambda_1)^{-1}\Lambda_1\right) W_1,$$

$$m_i = W_1^\top \Lambda_1 V \hat{B}_{t,i} = W_1^\top \Lambda_1(\hat{G}_{t,i} + \Lambda_1)^{-1}\hat{B}_{t,i}. \tag{21}$$

But notice that $V = (\hat{G}_{t,i} + \Lambda_1)^{-1} = \hat{\Sigma}_{t,i}$ and thus

$$\Omega_i = W_1^\top\left(\Lambda_1 - \Lambda_1 \hat{\Sigma}_{t,i}\Lambda_1\right)W_1, \qquad\qquad m_i = W_1^\top \Lambda_1 \hat{\Sigma}_{t,i}\hat{B}_{t,i}. \tag{22}$$

Finally, we plug this result in Eq. (20) to get

$$\mathbb{P}\left(H_t \mid \psi_{*,1} = \psi_1\right) = \prod_{i \in [K]} h_i(\psi_1) \propto \prod_{i \in [K]} \exp\left[-\frac{1}{2}\psi_1^\top \Omega_i \psi_1 + \psi_1^\top m_i\right],$$

$$= \exp\left[-\frac{1}{2}\psi_1^\top \sum_{i \in [K]} \Omega_i \psi_1 + \psi_1^\top \sum_{i \in [K]} m_i\right],$$

$$= \exp\left[-\frac{1}{2}\psi_1^\top \bar{G}_{t,1}\psi_1 + \psi_1^\top \bar{B}_{t,1}\right],$$

where

$$\bar{G}_{t,1} = \sum_{i=1}^K \Omega_i = \sum_{i=1}^K W_1^\top\left(\Lambda_1 - \Lambda_1 \hat{\Sigma}_{t,i}\Lambda_1\right)W_1 = W_1^\top \sum_{i=1}^K \left(\Sigma_1^{-1} - \Sigma_1^{-1}\hat{\Sigma}_{t,i}\Sigma_1^{-1}\right)W_1,$$

$$\bar{B}_{t,1} = \sum_{i=1}^K m_i = \sum_{i=1}^K \hat{\Sigma}_{t,i}\hat{B}_{t,i} = W_1^\top \Sigma_1^{-1}\sum_{i=1}^K \hat{\Sigma}_{t,i}\hat{B}_{t,i}.$$

This concludes the proof of the base case.

**(II) Induction step.** Let $\ell \in [L]/\{1\}$. Suppose that

$$\mathbb{P}\left(H_t \mid \psi_{*,\ell-1} = \psi_{\ell-1}\right) \propto \exp\left[-\frac{1}{2}\psi_{\ell-1}^\top \bar{G}_{t,\ell-1}\psi_{\ell-1} + \psi_{\ell-1}^\top \bar{B}_{t,\ell-1}\right]. \tag{23}$$

Then we want to show that

$$\mathbb{P}\left(H_t \mid \psi_{*,\ell} = \psi_\ell\right) \propto \exp\left[-\frac{1}{2}\psi_\ell^\top \bar{G}_{t,\ell}\psi_\ell + \psi_\ell^\top \bar{B}_{t,\ell}\right],$$

where

$$\bar{G}_{t,\ell} = W_\ell^\top\left(\Sigma_\ell^{-1} - \Sigma_\ell^{-1}\bar{\Sigma}_{t,\ell-1}\Sigma_\ell^{-1}\right)W_\ell = W_\ell^\top\left(\Sigma_\ell^{-1} - \Sigma_\ell^{-1}(\Sigma_\ell^{-1} + \bar{G}_{t,\ell-1})^{-1}\Sigma_\ell^{-1}\right)W_\ell,$$
$$\bar{B}_{t,\ell} = W_\ell^\top\Sigma_\ell^{-1}\bar{\Sigma}_{t,\ell-1}\bar{B}_{t,\ell-1} = W_\ell^\top\Sigma_\ell^{-1}(\Sigma_\ell^{-1} + \bar{G}_{t,\ell-1})^{-1}\bar{B}_{t,\ell-1}.$$

To achieve this, we start by expressing $\mathbb{P}\left(H_t \mid \psi_{*,\ell} = \psi_\ell\right)$ in terms of $\mathbb{P}\left(H_t \mid \psi_{*,\ell-1} = \psi_{\ell-1}\right)$ as

$$\mathbb{P}\left(H_t \mid \psi_{*,\ell} = \psi_\ell\right) = \int_{\psi_{\ell-1}} \mathbb{P}\left(H_t, \psi_{*,\ell-1} = \psi_{\ell-1} \mid \psi_{*,\ell} = \psi_\ell\right) \mathrm{d}\psi_{\ell-1},$$

$$= \int_{\psi_{\ell-1}} \mathbb{P}\left(H_t \mid \psi_{*,\ell-1} = \psi_{\ell-1}, \psi_{*,\ell} = \psi_\ell\right)\mathcal{N}(\psi_{\ell-1}; W_\ell\psi_\ell, \Sigma_\ell)\,\mathrm{d}\psi_{\ell-1},$$

$$= \int_{\psi_{\ell-1}} \mathbb{P}\left(H_t \mid \psi_{*,\ell-1} = \psi_{\ell-1}\right)\mathcal{N}(\psi_{\ell-1}; W_\ell\psi_\ell, \Sigma_\ell)\,\mathrm{d}\psi_{\ell-1},$$

$$\propto \int_{\psi_{\ell-1}} \exp\left[-\frac{1}{2}\psi_{\ell-1}^\top \bar{G}_{t,\ell-1}\psi_{\ell-1} + \psi_{\ell-1}^\top \bar{B}_{t,\ell-1}\right]\mathcal{N}(\psi_{\ell-1}; W_\ell\psi_\ell, \Sigma_\ell)\,\mathrm{d}\psi_{\ell-1},$$

$$\propto \int_{\psi_{\ell-1}} \exp\Big[-\frac{1}{2}\psi_{\ell-1}^\top \bar{G}_{t,\ell-1}\psi_{\ell-1} + \psi_{\ell-1}^\top \bar{B}_{t,\ell-1}$$
$$+ (\psi_{\ell-1} - W_\ell\psi_\ell)^\top \Lambda_\ell(\psi_{\ell-1} - W_\ell\psi_\ell)\Big]\,\mathrm{d}\psi_{\ell-1}.$$

Now let $S = \bar{G}_{t,\ell-1} + \Lambda_\ell$ and $V = \bar{B}_{t,\ell-1} + \Lambda_\ell W_\ell\psi_\ell$. Then we have that,

$$\mathbb{P}\left(H_t \mid \psi_{*,\ell} = \psi_\ell\right)$$
$$\propto \int_{\psi_{\ell-1}} \exp\Big[-\frac{1}{2}\psi_{\ell-1}^\top \bar{G}_{t,\ell-1}\psi_{\ell-1} + \psi_{\ell-1}^\top \bar{B}_{t,\ell-1}$$
$$+ (\psi_{\ell-1} - W_\ell\psi_\ell)^\top \Lambda_\ell(\psi_{\ell-1} - W_\ell\psi_\ell)\Big]\,\mathrm{d}\psi_{\ell-1},$$
$$\propto \int_{\psi_{\ell-1}} \exp\left[-\frac{1}{2}\left(\psi_{\ell-1}^\top S\psi_{\ell-1} - 2\psi_{\ell-1}^\top\left(\bar{B}_{t,\ell-1} + \Lambda_\ell W_\ell\psi_\ell\right) + \psi_\ell^\top W_\ell^\top \Lambda_\ell W_\ell\psi_\ell\right)\right]\mathrm{d}\psi_{\ell-1},$$
$$= \int_{\psi_{\ell-1}} \exp\left[-\frac{1}{2}\left(\psi_{\ell-1}^\top S(\psi_{\ell-1} - 2S^{-1}V) + \psi_\ell^\top W_\ell^\top \Lambda_\ell W_\ell\psi_\ell\right)\right]\mathrm{d}\psi_{\ell-1},$$
$$= \int_{\psi_{\ell-1}} \exp\Big[-\frac{1}{2}\Big((\psi_{\ell-1} - S^{-1}V)^\top S(\psi_{\ell-1} - S^{-1}V)$$
$$+ \psi_\ell^\top W_\ell^\top \Lambda_\ell W_\ell\psi_\ell - V^\top S^{-1}V\Big)\Big]\,\mathrm{d}\psi_{\ell-1}.$$

In the second step, we omit constants in $\psi_\ell$ and $\psi_{\ell-1}$. Thus

$$\mathbb{P}\left(H_t \mid \psi_{*,\ell} = \psi_\ell\right)$$
$$\propto \int_{\psi_{\ell-1}} \exp\left[-\frac{1}{2}\left((\psi_{\ell-1} - S^{-1}V)^\top S(\psi_{\ell-1} - S^{-1}V) + \psi_\ell^\top W_\ell^\top \Lambda_\ell W_\ell\psi_\ell - V^\top S^{-1}V\right)\right]\mathrm{d}\psi_{\ell-1},$$
$$\propto \exp\left[-\frac{1}{2}\left(\psi_\ell^\top W_\ell^\top \Lambda_\ell W_\ell\psi_\ell - V^\top S^{-1}V\right)\right].$$

It follows that

$$
\begin{aligned}
&\mathbb{P}\left(H_t \mid \psi_{*,\ell} = \psi_\ell\right) \\
&\propto \exp\left[-\frac{1}{2}\left(\psi_\ell^\top W_\ell^\top \Lambda_\ell W_\ell \psi_\ell - V^\top S^{-1} V\right)\right], \\
&= \exp\left[-\frac{1}{2}\left(\psi_\ell^\top W_\ell^\top \Lambda_\ell W_\ell \psi_\ell - \left(\bar{B}_{t,\ell-1} + \Lambda_\ell W_\ell \psi_\ell\right)^\top S^{-1}\left(\bar{B}_{t,\ell-1} + \Lambda_\ell W_\ell \psi_\ell\right)\right)\right] \\
&\propto \exp\left[-\frac{1}{2}\left(\psi_\ell^\top\left(W_\ell^\top \Lambda_\ell W_\ell - W_\ell^\top \Lambda_\ell S^{-1}\Lambda_\ell W_\ell\right)\psi_\ell - 2\psi_\ell^\top W_\ell^\top \Lambda_\ell S^{-1}\bar{B}_{t,\ell-1}\right)\right], \\
&= \exp\left[-\frac{1}{2}\psi_\ell^\top \bar{G}_{t,\ell}\psi_\ell + \psi_\ell^\top \bar{B}_{t,\ell}\right].
\end{aligned}
$$

In the last step, we omit constants in $\psi_\ell$ and we set

$$
\begin{aligned}
\bar{G}_{t,\ell} &= W_\ell^\top\left(\Lambda_\ell - \Lambda_\ell S^{-1}\Lambda_\ell\right)W_\ell = W_\ell^\top\left(\Lambda_\ell - \Lambda_\ell(\Lambda_\ell + \bar{G}_{t,\ell-1})^{-1}\Sigma_\ell^{-1}\Lambda_\ell\right)W_\ell, \\
\bar{B}_{t,\ell} &= W_\ell^\top \Lambda_\ell S^{-1}\bar{B}_{t,\ell-1} = W_\ell^\top \Lambda_\ell(\Lambda_\ell + \bar{G}_{t,\ell-1})^{-1}\bar{B}_{t,\ell-1}.
\end{aligned}
$$

This completes the proof. $\qquad\square$

## C  REGRET PROOF AND DISCUSSION

### C.1  TECHNICAL CONTRIBUTIONS

Our main technical contributions are the following.

**Lemma 2.** In `dTS`, sampling is done hierarchically, meaning the marginal posterior distribution of $\theta_{*,i}|H_t$ is not explicitly defined. Instead, we use the conditional posterior distribution of $\theta_{*,i}|H_t, \psi_{*,1}$. The first contribution was deriving $\theta_{*,i}|H_t$ using the total covariance decomposition combined with an induction proof, as our posteriors in Section 3.1 were derived recursively. Unlike in Bayes regret analysis for standard Thompson sampling, where the posterior distribution of $\theta_{*,i}|H_t$ is predetermined due to the absence of latent parameters, our method necessitates this recursive total covariance decomposition, marking a first difference from the standard Bayesian proofs of Thompson sampling. Note that `HierTS`, which is developed for multi-task linear bandits, also employs total covariance decomposition, but it does so under the assumption of a single latent parameter; on which action parameters are centered. Our extension significantly differs as it is tailored for contextual bandits with multiple, successive levels of latent parameters, moving away from `HierTS`'s assumption of a 1-level structure. Roughly speaking, `HierTS` when applied to contextual would consider a single-level hierarchy, where $\theta_{*,i}|\psi_{*,1} \sim \mathcal{N}(\psi_{*,1}, \Sigma_1)$ with $L = 1$. In contrast, our model proposes a multi-level hierarchy, where the first level is $\theta_{*,i}|\psi_{*,1} \sim \mathcal{N}(W_1\psi_{*,1}, \Sigma_1)$. This also introduces a new aspect to our approach – the use of a linear function $W_1\psi_{*,1}$, as opposed to `HierTS`'s assumption where action parameters are centered directly on the latent parameter. Thus, while `HierTS` also uses the total covariance decomposition, our generalize it to multi-level hierarchies under $L$ linear functions $W_\ell\psi_{*,\ell}$, instead of a single-level hierarchy under a single identity function $\psi_{*,1}$.

**Lemma 3.** In Bayes regret proofs for standard Thompson sampling, we often quantify the posterior information gain. This is achieved by monitoring the increase in posterior precision for the action taken $A_t$ in each round $t \in [n]$. However, in `dTS`, our analysis extends beyond this. We not only quantify the posterior information gain for the taken action but also for every latent parameter, since they are also learned. Lemma 3 addresses this aspect. To elaborate, we use the recursive formulas in Section 3.1 that connect the posterior covariance of each latent parameter $\psi_{*,\ell}$ with the covariance of the posterior action parameters $\theta_{*,i}$. This allows us to propagate the information gain associated with the action taken in round $A_t$ to all latent parameters $\psi_{*,\ell}$, for $\ell \in [L]$ by induction. This is a novel contribution, as it is not a feature of Bayes regret analyses in standard Thompson sampling.

**Proposition 1.** Building upon the insights of Theorem 1, we introduce the sparsity assumption (**A3**). Under this assumption, we demonstrate that the Bayes regret outlined in Theorem 1 can be significantly refined. Specifically, the regret becomes contingent on dimensions $d_\ell \le d$, as opposed to relying on the entire dimension $d$. This sparsity assumption is both a novel and a key technical contribution to our work. Its underlying principle is straightforward: the Bayes regret is influenced

by the quantity of parameters that require learning. With the sparsity assumption, this number is reduced to less than $d$ for each latent parameter. To substantiate this claim, we revisit the proof of Theorem 1 and modify a crucial equality. This adjustment results in a more precise representation by partitioning the covariance matrix of each latent parameter $\psi_{,\ell}$ into blocks. These blocks comprise a $d_\ell \times d_\ell$ segment corresponding to the learnable $d_\ell$ parameters of $\psi_{,\ell}$, and another block of size $(d - d_\ell) \times (d - d_\ell)$ that does not necessitate learning. This decomposition allows us to conclude that the final regret is solely dependent on $d_\ell$, marking a significant refinement from the original theorem.

## C.2 Proof of Lemma 2

In this proof, we heavily rely on the total covariance decomposition (Weiss, 2005). Also, refer to (Hong et al., 2022b, Section 5.2) for a brief introduction to this decomposition. Now, from Eq. (5), we have that

$$\mathrm{cov}\left[\theta_{*,i} \mid H_t, \psi_{*,1}\right] = \hat{\Sigma}_{t,i} = \left(\hat{G}_{t,i} + \Sigma_1^{-1}\right)^{-1},$$

$$\mathbb{E}\left[\theta_{*,i} \mid H_t, \psi_{*,1}\right] = \hat{\mu}_{t,i} = \hat{\Sigma}_{t,i}\left(\hat{B}_{t,i} + \Sigma_1^{-1}\mathrm{W}_1\psi_{*,1}\right).$$

First, given $H_t$, $\mathrm{cov}\left[\theta_{*,i} \mid H_t, \psi_{*,1}\right] = \left(\hat{G}_{t,i} + \Sigma_1^{-1}\right)^{-1}$ is constant. Thus

$$\mathbb{E}\left[\mathrm{cov}\left[\theta_{*,i} \mid H_t, \psi_{*,1}\right] \mid H_t\right] = \mathrm{cov}\left[\theta_{*,i} \mid H_t, \psi_{*,1}\right] = \left(\hat{G}_{t,i} + \Sigma_1^{-1}\right)^{-1} = \hat{\Sigma}_{t,i}.$$

In addition, given $H_t$, both $\hat{\Sigma}_{t,i}$ and $\hat{B}_{t,i}$ are constant. Thus

$$\mathrm{cov}\left[\mathbb{E}\left[\theta_{*,i} \mid H_t, \psi_{*,1}\right] \mid H_t\right] = \mathrm{cov}\left[\hat{\Sigma}_{t,i}\left(\hat{B}_{t,i} + \Sigma_1^{-1}\mathrm{W}_1\psi_{*,1}\right) \middle| H_t\right],$$

$$= \mathrm{cov}\left[\hat{\Sigma}_{t,i}\Sigma_1^{-1}\mathrm{W}_1\psi_{*,1} \middle| H_t\right],$$

$$= \hat{\Sigma}_{t,i}\Sigma_1^{-1}\mathrm{W}_1\mathrm{cov}\left[\psi_{*,1} \mid H_t\right]\mathrm{W}_1^\top\Sigma_1^{-1}\hat{\Sigma}_{t,i},$$

$$= \hat{\Sigma}_{t,i}\Sigma_1^{-1}\mathrm{W}_1\bar{\bar{\Sigma}}_{t,1}\mathrm{W}_1^\top\Sigma_1^{-1}\hat{\Sigma}_{t,i},$$

where $\bar{\bar{\Sigma}}_{t,1} = \mathrm{cov}\left[\psi_{*,1} \mid H_t\right]$ is the marginal posterior covariance of $\psi_{*,1}$. Finally, the total covariance decomposition (Weiss, 2005; Hong et al., 2022b) yields that

$$\check{\Sigma}_{t,i} = \mathrm{cov}\left[\theta_{*,i} \mid H_t\right] = \mathbb{E}\left[\mathrm{cov}\left[\theta_{*,i} \mid H_t, \psi_{*,1}\right] \mid H_t\right] + \mathrm{cov}\left[\mathbb{E}\left[\theta_{*,i} \mid H_t, \psi_{*,1}\right] \mid H_t\right],$$

$$= \hat{\Sigma}_{t,i} + \hat{\Sigma}_{t,i}\Sigma_1^{-1}\mathrm{W}_1\bar{\bar{\Sigma}}_{t,1}\mathrm{W}_1^\top\Sigma_1^{-1}\hat{\Sigma}_{t,i}, \tag{24}$$

However, $\bar{\bar{\Sigma}}_{t,1} = \mathrm{cov}\left[\psi_{*,1} \mid H_t\right]$ is different from $\bar{\Sigma}_{t,1} = \mathrm{cov}\left[\psi_{*,1} \mid H_t, \psi_{*,2}\right]$ that we already derived in Eq. (6). Thus we do not know the expression of $\bar{\bar{\Sigma}}_{t,1}$. But we can use the same total covariance decomposition trick to find it. Precisely, let $\bar{\bar{\Sigma}}_{t,\ell} = \mathrm{cov}\left[\psi_{*,\ell} \mid H_t\right]$ for any $\ell \in [L]$. Then we have that

$$\bar{\Sigma}_{t,1} = \mathrm{cov}\left[\psi_{*,1} \mid H_t, \psi_{*,2}\right] = \left(\Sigma_2^{-1} + \bar{G}_{t,1}\right)^{-1},$$

$$\bar{\mu}_{t,1} = \mathbb{E}\left[\psi_{*,1} \mid H_t, \psi_{*,2}\right] = \bar{\Sigma}_{t,1}\left(\Sigma_2^{-1}\mathrm{W}_2\psi_{*,2} + \bar{B}_{t,1}\right).$$

First, given $H_t$, $\mathrm{cov}\left[\psi_{*,1} \mid H_t, \psi_{*,2}\right] = \left(\Sigma_2^{-1} + \bar{G}_{t,1}\right)^{-1}$ is constant. Thus

$$\mathbb{E}\left[\mathrm{cov}\left[\psi_{*,1} \mid H_t, \psi_{*,2}\right] \mid H_t\right] = \mathrm{cov}\left[\psi_{*,1} \mid H_t, \psi_{*,2}\right] = \bar{\Sigma}_{t,1}.$$

In addition, given $H_t$, $\bar{\Sigma}_{t,1}$, $\tilde{\Sigma}_{t,1}$ and $\bar{B}_{t,1}$ are constant. Thus

$$\mathrm{cov}\left[\mathbb{E}\left[\psi_{*,1} \mid H_t, \psi_{*,2}\right] \mid H_t\right] = \mathrm{cov}\left[\bar{\Sigma}_{t,1}\left(\Sigma_2^{-1}\mathrm{W}_2\psi_{*,2} + \bar{B}_{t,1}\right) \middle| H_t\right],$$

$$= \mathrm{cov}\left[\bar{\Sigma}_{t,1}\Sigma_2^{-1}\mathrm{W}_2\psi_{*,2} \middle| H_t\right],$$

$$= \bar{\Sigma}_{t,1}\Sigma_2^{-1}\mathrm{W}_2\mathrm{cov}\left[\psi_{*,2} \mid H_t\right]\mathrm{W}_2^\top\Sigma_2^{-1}\bar{\Sigma}_{t,1},$$

$$= \bar{\Sigma}_{t,1}\Sigma_2^{-1}\mathrm{W}_2\bar{\bar{\Sigma}}_{t,2}\mathrm{W}_2^\top\Sigma_2^{-1}\bar{\Sigma}_{t,1}.$$

Finally, total covariance decomposition (Weiss, 2005; Hong et al., 2022b) leads to

$$\bar{\bar{\Sigma}}_{t,1} = \mathrm{cov}\left[\psi_{*,1} \,|\, H_t\right] = \mathbb{E}\left[\mathrm{cov}\left[\psi_{*,1} \,|\, H_t, \psi_{*,2}\right] | H_t\right] + \mathrm{cov}\left[\mathbb{E}\left[\psi_{*,1} \,|\, H_t, \psi_{*,2}\right] | H_t\right],$$

$$= \bar{\Sigma}_{t,1} + \bar{\Sigma}_{t,1}\Sigma_2^{-1}\mathrm{W}_2\bar{\bar{\Sigma}}_{t,2}\mathrm{W}_2^\top\Sigma_2^{-1}\bar{\Sigma}_{t,1}.$$

Now using the techniques, this can be generalized using the same technique as above to

$$\bar{\bar{\Sigma}}_{t,\ell} = \bar{\Sigma}_{t,\ell} + \bar{\Sigma}_{t,\ell}\Sigma_{\ell+1}^{-1}\mathrm{W}_{\ell+1}\bar{\bar{\Sigma}}_{t,\ell+1}\mathrm{W}_{\ell+1}^\top\Sigma_{\ell+1}^{-1}\bar{\Sigma}_{t,\ell}, \qquad \forall\ell \in [L-1].$$

Then, by induction, we get that

$$\bar{\bar{\Sigma}}_{t,1} = \sum_{\ell\in[L]} \bar{\mathrm{P}}_\ell\bar{\Sigma}_{t,\ell}\bar{\mathrm{P}}_\ell^\top, \qquad \forall\ell \in [L-1],$$

where we use that by definition $\bar{\bar{\Sigma}}_{t,L} = \mathrm{cov}\left[\psi_{*,L} \,|\, H_t\right] = \bar{\Sigma}_{t,L}$ and set $\bar{\mathrm{P}}_1 = I_d$ and $\bar{\mathrm{P}}_\ell = \prod_{k=1}^{\ell-1}\bar{\Sigma}_{t,k}\Sigma_{k+1}^{-1}\mathrm{W}_{k+1}$ for any $\ell \in [L]/\{1\}$. Plugging this in Eq. (24) leads to

$$\check{\Sigma}_{t,i} = \hat{\Sigma}_{t,i} + \sum_{\ell\in[L]} \hat{\Sigma}_{t,i}\Sigma_1^{-1}\mathrm{W}_1\bar{\mathrm{P}}_\ell\bar{\Sigma}_{t,\ell}\bar{\mathrm{P}}_\ell^\top\mathrm{W}_1^\top\Sigma_1^{-1}\hat{\Sigma}_{t,i},$$

$$= \hat{\Sigma}_{t,i} + \sum_{\ell\in[L]} \hat{\Sigma}_{t,i}\Sigma_1^{-1}\mathrm{W}_1\bar{\mathrm{P}}_\ell\bar{\Sigma}_{t,\ell}(\hat{\Sigma}_{t,i}\Sigma_1^{-1}\mathrm{W}_1)^\top,$$

$$= \hat{\Sigma}_{t,i} + \sum_{\ell\in[L]} \mathrm{P}_{i,\ell}\bar{\Sigma}_{t,\ell}\mathrm{P}_{i,\ell}^\top,$$

where $\mathrm{P}_{i,\ell} = \hat{\Sigma}_{t,i}\Sigma_1^{-1}\mathrm{W}_1\bar{\mathrm{P}}_\ell = \hat{\Sigma}_{t,i}\Sigma_1^{-1}\mathrm{W}_1\prod_{k=1}^{\ell-1}\bar{\Sigma}_{t,k}\Sigma_{k+1}^{-1}\mathrm{W}_{k+1}$.

## C.3 PROOF OF LEMMA 3

We prove this result by induction. We start with the base case when $\ell = 1$.

**(I) Base case.** Let $u = \sigma^{-1}\hat{\Sigma}_{t,A_t}^{\frac{1}{2}}X_t$ From the expression of $\bar{\Sigma}_{t,1}$ in Eq. (6), we have that

$$\bar{\Sigma}_{t+1,1}^{-1} - \bar{\Sigma}_{t,1}^{-1} = \mathrm{W}_1^\top\left(\Sigma_1^{-1} - \Sigma_1^{-1}(\hat{\Sigma}_{t,A_t}^{-1} + \sigma^{-2}X_tX_t^\top)^{-1}\Sigma_1^{-1} - (\Sigma_1^{-1} - \Sigma_1^{-1}\hat{\Sigma}_{t,A_t}\Sigma_1^{-1})\right)\mathrm{W}_1,$$

$$= \mathrm{W}_1^\top\left(\Sigma_1^{-1}(\hat{\Sigma}_{t,A_t} - (\hat{\Sigma}_{t,A_t}^{-1} + \sigma^{-2}X_tX_t^\top)^{-1})\Sigma_1^{-1}\right)\mathrm{W}_1,$$

$$= \mathrm{W}_1^\top\left(\Sigma_1^{-1}\hat{\Sigma}_{t,A_t}^{\frac{1}{2}}(I_d - (I_d + \sigma^{-2}\hat{\Sigma}_{t,A_t}^{\frac{1}{2}}X_tX_t^\top\hat{\Sigma}_{t,A_t}^{\frac{1}{2}})^{-1})\hat{\Sigma}_{t,A_t}^{\frac{1}{2}}\Sigma_1^{-1}\right)\mathrm{W}_1,$$

$$= \mathrm{W}_1^\top\left(\Sigma_1^{-1}\hat{\Sigma}_{t,A_t}^{\frac{1}{2}}(I_d - (I_d + uu^\top)^{-1})\hat{\Sigma}_{t,A_t}^{\frac{1}{2}}\Sigma_1^{-1}\right)\mathrm{W}_1,$$

$$\stackrel{(i)}{=} \mathrm{W}_1^\top\left(\Sigma_1^{-1}\hat{\Sigma}_{t,A_t}^{\frac{1}{2}}\frac{uu^\top}{1+u^\top u}\hat{\Sigma}_{t,A_t}^{\frac{1}{2}}\Sigma_1^{-1}\right)\mathrm{W}_1,$$

$$\stackrel{(ii)}{=} \sigma^{-2}\mathrm{W}_1^\top\Sigma_1^{-1}\hat{\Sigma}_{t,A_t}\frac{X_tX_t^\top}{1+u^\top u}\hat{\Sigma}_{t,A_t}\Sigma_1^{-1}\mathrm{W}_1. \tag{25}$$

In $(i)$ we use the Sherman-Morrison formula. Note that $(ii)$ says that $\bar{\Sigma}_{t+1,1}^{-1} - \bar{\Sigma}_{t,1}^{-1}$ is one-rank which we will also need in induction step. Now, we have that $\|X_t\|^2 = 1$. Therefore,

$$1 + u^\top u = 1 + \sigma^{-2}X_t^\top\hat{\Sigma}_{t,A_t}X_t \le 1 + \sigma^{-2}\lambda_1(\Sigma_1)\|X_t\|^2 = 1 + \sigma^{-2}\sigma_1^2 \le \sigma_{\mathrm{MAX}}^2,$$

where we use that by definition of $\sigma_{\mathrm{MAX}}^2$ in Lemma 3, we have that $\sigma_{\mathrm{MAX}}^2 \ge 1 + \sigma^{-2}\sigma_1^2$. Therefore, by taking the inverse, we get that $\frac{1}{1+u^\top u} \ge \sigma_{\mathrm{MAX}}^{-2}$. Combining this with Eq. (25) leads to

$$\bar{\Sigma}_{t+1,1}^{-1} - \bar{\Sigma}_{t,1}^{-1} \succeq \sigma^{-2}\sigma_{\mathrm{MAX}}^{-2}\mathrm{W}_1^\top\Sigma_1^{-1}\hat{\Sigma}_{t,A_t}X_tX_t^\top\hat{\Sigma}_{t,A_t}\Sigma_1^{-1}\mathrm{W}_1$$

Noticing that $\mathrm{P}_{A_t,1} = \hat{\Sigma}_{t,A_t}\Sigma_1^{-1}\mathrm{W}_1$ concludes the proof of the base case when $\ell = 1$.

**(II) Induction step.** Let $\ell \in [L]/\{1\}$ and suppose that $\bar{\Sigma}_{t+1,\ell-1}^{-1} - \bar{\Sigma}_{t,\ell-1}^{-1}$ is one-rank and that it holds for $\ell - 1$ that

$$\bar{\Sigma}_{t+1,\ell-1}^{-1} - \bar{\Sigma}_{t,\ell-1}^{-1} \succeq \sigma^{-2}\sigma_{\mathrm{MAX}}^{-2(\ell-1)}\mathrm{P}_{A_t,\ell-1}^\top X_tX_t^\top\mathrm{P}_{A_t,\ell-1}, \quad \text{where } \sigma_{\mathrm{MAX}}^{-2} = \max_{\ell\in[L]} 1 + \sigma^{-2}\sigma_\ell^2.$$

Then, we want to show that $\bar{\Sigma}_{t+1,\ell}^{-1} - \bar{\Sigma}_{t,\ell}^{-1}$ is also one-rank and that it holds that

$$\bar{\Sigma}_{t+1,\ell}^{-1} - \bar{\Sigma}_{t,\ell}^{-1} \succeq \sigma^{-2} \sigma_{\text{MAX}}^{-2\ell} \mathrm{P}_{A_t,\ell}^\top X_t X_t^\top \mathrm{P}_{A_t,\ell}, \qquad \text{where } \sigma_{\text{MAX}}^{-2} = \max_{\ell \in [L]} 1 + \sigma^{-2} \sigma_\ell^2.$$

This is achieved as follows. First, we notice that by the induction hypothesis, we have that $\tilde{\Sigma}_{t+1,\ell-1}^{-1} - \bar{G}_{t,\ell-1} = \bar{\Sigma}_{t+1,\ell-1}^{-1} - \bar{\Sigma}_{t,\ell-1}^{-1}$ is one-rank. In addition, the matrix is positive semi-definite. Thus we can write it as $\tilde{\Sigma}_{t+1,\ell-1}^{-1} - \bar{G}_{t,\ell-1} = uu^\top$ where $u \in \mathbb{R}^d$. Then, similarly to the base case, we have

$$
\begin{aligned}
\bar{\Sigma}_{t+1,\ell}^{-1} - \bar{\Sigma}_{t,\ell}^{-1} &= \tilde{\Sigma}_{t+1,\ell}^{-1} - \tilde{\Sigma}_{t,\ell}^{-1}, \\
&= \mathrm{W}_\ell^\top \left( \Sigma_\ell + \tilde{\Sigma}_{t+1,\ell-1} \right)^{-1} \mathrm{W}_\ell - \mathrm{W}_\ell^\top \left( \Sigma_\ell + \tilde{\Sigma}_{t,\ell-1} \right)^{-1} \mathrm{W}_\ell, \\
&= \mathrm{W}_\ell^\top \left[ \left( \Sigma_\ell + \tilde{\Sigma}_{t+1,\ell-1} \right)^{-1} - \left( \Sigma_\ell + \tilde{\Sigma}_{t,\ell-1} \right)^{-1} \right] \mathrm{W}_\ell, \\
&= \mathrm{W}_\ell^\top \Sigma_\ell^{-1} \left[ \left( \Sigma_\ell^{-1} + \bar{G}_{t,\ell-1} \right)^{-1} - \left( \Sigma_\ell^{-1} + \tilde{\Sigma}_{t+1,\ell-1}^{-1} \right)^{-1} \right] \Sigma_\ell^{-1} \mathrm{W}_\ell, \\
&= \mathrm{W}_\ell^\top \Sigma_\ell^{-1} \left[ \left( \Sigma_\ell^{-1} + \bar{G}_{t,\ell-1} \right)^{-1} - \left( \Sigma_\ell^{-1} + \bar{G}_{t,\ell-1} + \tilde{\Sigma}_{t+1,\ell-1}^{-1} - \bar{G}_{t,\ell-1} \right)^{-1} \right] \Sigma_\ell^{-1} \mathrm{W}_\ell, \\
&= \mathrm{W}_\ell^\top \Sigma_\ell^{-1} \left[ \left( \Sigma_\ell^{-1} + \bar{G}_{t,\ell-1} \right)^{-1} - \left( \Sigma_\ell^{-1} + \bar{G}_{t,\ell-1} + uu^\top \right)^{-1} \right] \Sigma_\ell^{-1} \mathrm{W}_\ell, \\
&= \mathrm{W}_\ell^\top \Sigma_\ell^{-1} \left[ \bar{\Sigma}_{t,\ell-1} - \left( \bar{\Sigma}_{t,\ell-1}^{-1} + uu^\top \right)^{-1} \right] \Sigma_\ell^{-1} \mathrm{W}_\ell, \\
&= \mathrm{W}_\ell^\top \Sigma_\ell^{-1} \left[ \bar{\Sigma}_{t,\ell-1} \frac{uu^\top}{1 + u^\top \bar{\Sigma}_{t,\ell-1} u} \bar{\Sigma}_{t,\ell-1} \right] \Sigma_\ell^{-1} \mathrm{W}_\ell, \\
&= \mathrm{W}_\ell^\top \Sigma_\ell^{-1} \bar{\Sigma}_{t,\ell-1} \frac{uu^\top}{1 + u^\top \bar{\Sigma}_{t,\ell-1} u} \bar{\Sigma}_{t,\ell-1} \Sigma_\ell^{-1} \mathrm{W}_\ell
\end{aligned}
$$

However, we it follows from the induction hypothesis that $uu^\top = \tilde{\Sigma}_{t+1,\ell-1}^{-1} - \bar{G}_{t,\ell-1} = \bar{\Sigma}_{t+1,\ell-1}^{-1} - \bar{\Sigma}_{t,\ell-1}^{-1} \succeq \sigma^{-2} \sigma_{\text{MAX}}^{-2(\ell-1)} \mathrm{P}_{A_t,\ell-1}^\top X_t X_t^\top \mathrm{P}_{A_t,\ell-1}$. Therefore,

$$
\begin{aligned}
\bar{\Sigma}_{t+1,\ell}^{-1} - \bar{\Sigma}_{t,\ell}^{-1} &= \mathrm{W}_\ell^\top \Sigma_\ell^{-1} \bar{\Sigma}_{t,\ell-1} \frac{uu^\top}{1 + u^\top \bar{\Sigma}_{t,\ell-1} u} \bar{\Sigma}_{t,\ell-1} \Sigma_\ell^{-1} \mathrm{W}_\ell, \\
&\succeq \mathrm{W}_\ell^\top \Sigma_\ell^{-1} \bar{\Sigma}_{t,\ell-1} \frac{\sigma^{-2} \sigma_{\text{MAX}}^{-2(\ell-1)} \mathrm{P}_{A_t,\ell-1}^\top X_t X_t^\top \mathrm{P}_{A_t,\ell-1}}{1 + u^\top \bar{\Sigma}_{t,\ell-1} u} \bar{\Sigma}_{t,\ell-1} \Sigma_\ell^{-1} \mathrm{W}_\ell, \\
&= \frac{\sigma^{-2} \sigma_{\text{MAX}}^{-2(\ell-1)}}{1 + u^\top \bar{\Sigma}_{t,\ell-1} u} \mathrm{W}_\ell^\top \Sigma_\ell^{-1} \bar{\Sigma}_{t,\ell-1} \mathrm{P}_{A_t,\ell-1}^\top X_t X_t^\top \mathrm{P}_{A_t,\ell-1} \bar{\Sigma}_{t,\ell-1} \Sigma_\ell^{-1} \mathrm{W}_\ell, \\
&= \frac{\sigma^{-2} \sigma_{\text{MAX}}^{-2(\ell-1)}}{1 + u^\top \bar{\Sigma}_{t,\ell-1} u} \mathrm{P}_{A_t,\ell}^\top X_t X_t^\top \mathrm{P}_{A_t,\ell}.
\end{aligned}
$$

Finally, we use that $1 + u^\top \bar{\Sigma}_{t,\ell-1} u \leq 1 + \|u\|_2 \lambda_1(\bar{\Sigma}_{t,\ell-1}) \leq 1 + \sigma^{-2} \sigma_\ell^2$. Here we use that $\|u\|_2 \leq \sigma^{-2}$, which can also be proven by induction, and that $\lambda_1(\bar{\Sigma}_{t,\ell-1}) \leq \sigma_\ell^2$, which follows from the expression of $\bar{\Sigma}_{t,\ell-1}$ in Section 3.1. Therefore, we have that

$$
\begin{aligned}
\bar{\Sigma}_{t+1,\ell}^{-1} - \bar{\Sigma}_{t,\ell}^{-1} &\succeq \frac{\sigma^{-2} \sigma_{\text{MAX}}^{-2(\ell-1)}}{1 + u^\top \bar{\Sigma}_{t,\ell-1} u} \mathrm{P}_{A_t,\ell}^\top X_t X_t^\top \mathrm{P}_{A_t,\ell}, \\
&\succeq \frac{\sigma^{-2} \sigma_{\text{MAX}}^{-2(\ell-1)}}{1 + \sigma^{-2} \sigma_\ell^2} \mathrm{P}_{A_t,\ell}^\top X_t X_t^\top \mathrm{P}_{A_t,\ell}, \\
&\succeq \sigma^{-2} \sigma_{\text{MAX}}^{-2\ell} \mathrm{P}_{A_t,\ell}^\top X_t X_t^\top \mathrm{P}_{A_t,\ell},
\end{aligned}
$$

where the last inequality follows from the definition of $\sigma_{\text{MAX}}^2 = \max_{\ell \in [L]} 1 + \sigma^{-2} \sigma_\ell^2$. This concludes the proof.

### C.4 PROOF OF THEOREM 1

We start with the following standard result which we borrow from (Hong et al., 2022a; Aouali et al., 2023),

$$\mathcal{BR}(n) \leq \sqrt{2n \log(1/\delta)} \sqrt{\mathbb{E}\left[\sum_{t=1}^{n} \|X_t\|_{\check{\Sigma}_{t,A_t}}^2\right]} + cn\delta, \qquad \text{where } c > 0 \text{ is a constant}. \quad (26)$$

Then we use Lemma 2 and express the marginal covariance $\check{\Sigma}_{t,A_t}$ as

$$\check{\Sigma}_{t,i} = \hat{\Sigma}_{t,i} + \sum_{\ell \in [L]} \mathrm{P}_{i,\ell} \bar{\Sigma}_{t,\ell} \mathrm{P}_{i,\ell}^\top, \qquad \text{where } \mathrm{P}_{i,\ell} = \hat{\Sigma}_{t,i} \Sigma_1^{-1} \mathrm{W}_1 \prod_{k=1}^{\ell-1} \bar{\Sigma}_{t,k} \Sigma_{k+1}^{-1} \mathrm{W}_{k+1}. \quad (27)$$

Therefore, we can decompose $\|X_t\|_{\check{\Sigma}_{t,A_t}}^2$ as

$$\|X_t\|_{\check{\Sigma}_{t,A_t}}^2 = \sigma^2 \frac{X_t^\top \check{\Sigma}_{t,A_t} X_t}{\sigma^2} \stackrel{(i)}{=} \sigma^2 \big(\sigma^{-2} X_t^\top \hat{\Sigma}_{t,A_t} X_t + \sigma^{-2} \sum_{\ell \in [L]} X_t^\top \mathrm{P}_{A_t,\ell} \bar{\Sigma}_{t,\ell} \mathrm{P}_{A_t,\ell}^\top X_t \big),$$

$$\stackrel{(ii)}{\leq} c_0 \log(1 + \sigma^{-2} X_t^\top \hat{\Sigma}_{t,A_t} X_t) + \sum_{\ell \in [L]} c_\ell \log(1 + \sigma^{-2} X_t^\top \mathrm{P}_{A_t,\ell} \bar{\Sigma}_{t,\ell} \mathrm{P}_{A_t,\ell}^\top X_t), \quad (28)$$

where $(i)$ follows from Eq. (27), and we use the following inequality in $(ii)$

$$x = \frac{x}{\log(1+x)} \log(1+x) \leq \left(\max_{x \in [0,u]} \frac{x}{\log(1+x)}\right) \log(1+x) = \frac{u}{\log(1+u)} \log(1+x),$$

which holds for any $x \in [0, u]$, where constants $c_0$ and $c_\ell$ are derived as

$$c_0 = \frac{\sigma_1^2}{\log(1 + \frac{\sigma_1^2}{\sigma^2})}, \quad c_\ell = \frac{\sigma_{\ell+1}^2}{\log(1 + \frac{\sigma_{\ell+1}^2}{\sigma^2})}, \text{ with the convention that } \sigma_{L+1} = 1.$$

The derivation of $c_0$ uses that

$$X_t^\top \hat{\Sigma}_{t,A_t} X_t \leq \lambda_1(\hat{\Sigma}_{t,A_t}) \|X_t\|^2 \leq \lambda_d^{-1}(\Sigma_1^{-1} + G_{t,A_t}) \leq \lambda_d^{-1}(\Sigma_1^{-1}) = \lambda_1(\Sigma_1) = \sigma_1^2.$$

The derivation of $c_\ell$ follows from

$$X_t^\top \mathrm{P}_{A_t,\ell} \bar{\Sigma}_{t,\ell} \mathrm{P}_{A_t,\ell}^\top X_t \leq \lambda_1(\mathrm{P}_{A_t,\ell} \mathrm{P}_{A_t,\ell}^\top) \lambda_1(\bar{\Sigma}_{t,\ell}) \|X_t\|^2 \leq \sigma_{\ell+1}^2.$$

Therefore, from Eq. (28) and Eq. (26), we get that

$$\mathcal{BR}(n) \leq \sqrt{2n \log(1/\delta)} \Big(\mathbb{E}\Big[c_0 \sum_{t=1}^{n} \log(1 + \sigma^{-2} X_t^\top \hat{\Sigma}_{t,A_t} X_t)$$

$$+ \sum_{\ell \in [L]} c_\ell \sum_{t=1}^{n} \log(1 + \sigma^{-2} X_t^\top \mathrm{P}_{A_t,\ell} \bar{\Sigma}_{t,\ell} \mathrm{P}_{A_t,\ell}^\top X_t)\Big]\Big)^{\frac{1}{2}} + cn\delta \quad (29)$$

Now we focus on bounding the logarithmic terms in Eq. (29).

**(I) First term in Eq. (29)** We first rewrite this term as

$$\log(1 + \sigma^{-2} X_t^\top \hat{\Sigma}_{t,A_t} X_t) \stackrel{(i)}{=} \log \det(I_d + \sigma^{-2} \hat{\Sigma}_{t,A_t}^{\frac{1}{2}} X_t X_t^\top \hat{\Sigma}_{t,A_t}^{\frac{1}{2}}),$$

$$= \log \det(\hat{\Sigma}_{t,A_t}^{-1} + \sigma^{-2} X_t X_t^\top) - \log \det(\hat{\Sigma}_{t,A_t}^{-1}) = \log \det(\hat{\Sigma}_{t+1,A_t}^{-1}) - \log \det(\hat{\Sigma}_{t,A_t}^{-1}),$$

where $(i)$ follows from the Weinstein–Aronszajn identity. Then we sum over all rounds $t \in [n]$, and get a telescoping

$$\sum_{t=1}^{n} \log \det(I_d + \sigma^{-2} \hat{\Sigma}_{t,A_t}^{\frac{1}{2}} X_t X_t^\top \hat{\Sigma}_{t,A_t}^{\frac{1}{2}}) = \sum_{t=1}^{n} \log \det(\hat{\Sigma}_{t+1,A_t}^{-1}) - \log \det(\hat{\Sigma}_{t,A_t}^{-1}),$$

$$= \sum_{t=1}^{n} \sum_{i=1}^{K} \log \det(\hat{\Sigma}_{t+1,i}^{-1}) - \log \det(\hat{\Sigma}_{t,i}^{-1}) = \sum_{i=1}^{K} \sum_{t=1}^{n} \log \det(\hat{\Sigma}_{t+1,i}^{-1}) - \log \det(\hat{\Sigma}_{t,i}^{-1}),$$

$$= \sum_{i=1}^{K} \log \det(\hat{\Sigma}_{n+1,i}^{-1}) - \log \det(\hat{\Sigma}_{1,i}^{-1}) \stackrel{(i)}{=} \sum_{i=1}^{K} \log \det(\Sigma_1^{\frac{1}{2}} \hat{\Sigma}_{n+1,i}^{-1} \Sigma_1^{\frac{1}{2}}),$$

where $(i)$ follows from the fact that $\hat{\Sigma}_{1,i} = \Sigma_1$. Now we use the inequality of arithmetic and geometric means and get

$$
\begin{aligned}
\sum_{t=1}^{n} \log \det(I_d + \sigma^{-2}\hat{\Sigma}_{t,A_t}^{\frac{1}{2}} X_t X_t^{\top} \hat{\Sigma}_{t,A_t}^{\frac{1}{2}}) &= \sum_{i=1}^{K} \log \det(\Sigma_1^{\frac{1}{2}} \hat{\Sigma}_{n+1,i}^{-1} \Sigma_1^{\frac{1}{2}}), \\
&\leq \sum_{i=1}^{K} d \log \left( \frac{1}{d} \operatorname{Tr}(\Sigma_1^{\frac{1}{2}} \hat{\Sigma}_{n+1,i}^{-1} \Sigma_1^{\frac{1}{2}}) \right), \quad\quad (30) \\
&\leq \sum_{i=1}^{K} d \log \left( 1 + \frac{n}{d}\frac{\sigma_1^2}{\sigma^2} \right) = K d \log \left( 1 + \frac{n}{d}\frac{\sigma_1^2}{\sigma^2} \right).
\end{aligned}
$$

**(II) Remaining terms in Eq. (29)** Let $\ell \in [L]$. Then we have that

$$
\begin{aligned}
\log(1 + \sigma^{-2} X_t^{\top} \mathrm{P}_{A_t,\ell} \bar{\Sigma}_{t,\ell} \mathrm{P}_{A_t,\ell}^{\top} X_t) &= \sigma_{\mathrm{MAX}}^{2\ell} \sigma_{\mathrm{MAX}}^{-2\ell} \log(1 + \sigma^{-2} X_t^{\top} \mathrm{P}_{A_t,\ell} \bar{\Sigma}_{t,\ell} \mathrm{P}_{A_t,\ell}^{\top} X_t), \\
&\leq \sigma_{\mathrm{MAX}}^{2\ell} \log(1 + \sigma^{-2} \sigma_{\mathrm{MAX}}^{-2\ell} X_t^{\top} \mathrm{P}_{A_t,\ell} \bar{\Sigma}_{t,\ell} \mathrm{P}_{A_t,\ell}^{\top} X_t), \\
&\overset{(i)}{=} \sigma_{\mathrm{MAX}}^{2\ell} \log \det(I_d + \sigma^{-2} \sigma_{\mathrm{MAX}}^{-2\ell} \bar{\Sigma}_{t,\ell}^{\frac{1}{2}} \mathrm{P}_{A_t,\ell}^{\top} X_t X_t^{\top} \mathrm{P}_{A_t,\ell} \bar{\Sigma}_{t,\ell}^{\frac{1}{2}}), \\
&= \sigma_{\mathrm{MAX}}^{2\ell} \Big( \log \det(\bar{\Sigma}_{t,\ell}^{-1} + \sigma^{-2} \sigma_{\mathrm{MAX}}^{-2\ell} \mathrm{P}_{A_t,\ell}^{\top} X_t X_t^{\top} \mathrm{P}_{A_t,\ell}) - \log \det(\bar{\Sigma}_{t,\ell}^{-1}) \Big),
\end{aligned}
$$

where we use the Weinstein–Aronszajn identity in $(i)$. Now we know from Lemma 3 that the following inequality holds $\sigma^{-2} \sigma_{\mathrm{MAX}}^{-2\ell} \mathrm{P}_{A_t,\ell}^{\top} X_t X_t^{\top} \mathrm{P}_{A_t,\ell} \preceq \bar{\Sigma}_{t+1,\ell}^{-1} - \bar{\Sigma}_{t,\ell}^{-1}$. As a result, we get that $\bar{\Sigma}_{t,\ell}^{-1} + \sigma^{-2} \sigma_{\mathrm{MAX}}^{-2\ell} \mathrm{P}_{A_t,\ell}^{\top} X_t X_t^{\top} \mathrm{P}_{A_t,\ell} \preceq \bar{\Sigma}_{t+1,\ell}^{-1}$. Thus,

$$
\log(1 + \sigma^{-2} X_t^{\top} \mathrm{P}_{A_t,\ell} \bar{\Sigma}_{t,\ell} \mathrm{P}_{A_t,\ell}^{\top} X_t) \leq \sigma_{\mathrm{MAX}}^{2\ell} \Big( \log \det(\bar{\Sigma}_{t+1,\ell}^{-1}) - \log \det(\bar{\Sigma}_{t,\ell}^{-1}) \Big),
$$

Then we sum over all rounds $t \in [n]$, and get a telescoping

$$
\begin{aligned}
\sum_{t=1}^{n} \log(1 + \sigma^{-2} X_t^{\top} \mathrm{P}_{A_t,\ell} \bar{\Sigma}_{t,\ell} \mathrm{P}_{A_t,\ell}^{\top} X_t) &\leq \sigma_{\mathrm{MAX}}^{2\ell} \sum_{t=1}^{n} \log \det(\bar{\Sigma}_{t+1,\ell}^{-1}) - \log \det(\bar{\Sigma}_{t,\ell}^{-1}), \\
&= \sigma_{\mathrm{MAX}}^{2\ell} \Big( \log \det(\bar{\Sigma}_{n+1,\ell}^{-1}) - \log \det(\bar{\Sigma}_{1,\ell}^{-1}) \Big), \\
&\overset{(i)}{=} \sigma_{\mathrm{MAX}}^{2\ell} \Big( \log \det(\bar{\Sigma}_{n+1,\ell}^{-1}) - \log \det(\Sigma_{\ell+1}^{-1}) \Big), \\
&= \sigma_{\mathrm{MAX}}^{2\ell} \Big( \log \det(\Sigma_{\ell+1}^{\frac{1}{2}} \bar{\Sigma}_{n+1,\ell}^{-1} \Sigma_{\ell+1}^{\frac{1}{2}}) \Big),
\end{aligned}
$$

where we use that $\bar{\Sigma}_{1,\ell} = \Sigma_{\ell+1}$ in $(i)$. Finally, we use the inequality of arithmetic and geometric means and get that

$$
\begin{aligned}
\sum_{t=1}^{n} \log(1 + \sigma^{-2} X_t^{\top} \mathrm{P}_{A_t,\ell} \bar{\Sigma}_{t,\ell} \mathrm{P}_{A_t,\ell}^{\top} X_t) &\leq \sigma_{\mathrm{MAX}}^{2\ell} \Big( \log \det(\Sigma_{\ell+1}^{\frac{1}{2}} \bar{\Sigma}_{n+1,\ell}^{-1} \Sigma_{\ell+1}^{\frac{1}{2}}) \Big), \\
&\leq d\sigma_{\mathrm{MAX}}^{2\ell} \log \left( \frac{1}{d} \operatorname{Tr}(\Sigma_{\ell+1}^{\frac{1}{2}} \bar{\Sigma}_{n+1,\ell}^{-1} \Sigma_{\ell+1}^{\frac{1}{2}}) \right), \quad (31) \\
&\leq d\sigma_{\mathrm{MAX}}^{2\ell} \log \left( 1 + \frac{\sigma_{\ell+1}^2}{\sigma_\ell^2} \right),
\end{aligned}
$$

The last inequality follows from the expression of $\bar{\Sigma}_{n+1,\ell}^{-1}$ in Eq. (6) that leads to

$$
\begin{aligned}
\Sigma_{\ell+1}^{\frac{1}{2}} \bar{\Sigma}_{n+1,\ell}^{-1} \Sigma_{\ell+1}^{\frac{1}{2}} &= I_d + \Sigma_{\ell+1}^{\frac{1}{2}} \bar{G}_{t,\ell} \Sigma_{\ell+1}^{\frac{1}{2}}, \\
&= I_d + \Sigma_{\ell+1}^{\frac{1}{2}} \mathrm{W}_\ell^{\top} \big( \Sigma_\ell^{-1} - \Sigma_\ell^{-1} \bar{\Sigma}_{t,\ell-1} \Sigma_\ell^{-1} \big) \mathrm{W}_\ell \Sigma_{\ell+1}^{\frac{1}{2}}, \quad (32)
\end{aligned}
$$

since $\bar{G}_{t,\ell} = W_\ell^\top\left(\Sigma_\ell^{-1} - \Sigma_\ell^{-1}\bar{\Sigma}_{t,\ell-1}\Sigma_\ell^{-1}\right)W_\ell$. This allows us to bound $\frac{1}{d}\operatorname{Tr}(\Sigma_{\ell+1}^{\frac{1}{2}}\bar{\Sigma}_{n+1,\ell}^{-1}\Sigma_{\ell+1}^{\frac{1}{2}})$ as

$$
\begin{aligned}
\frac{1}{d}\operatorname{Tr}(\Sigma_{\ell+1}^{\frac{1}{2}}\bar{\Sigma}_{n+1,\ell}^{-1}\Sigma_{\ell+1}^{\frac{1}{2}}) &= \frac{1}{d}\operatorname{Tr}(I_d + \Sigma_{\ell+1}^{\frac{1}{2}}W_\ell^\top\left(\Sigma_\ell^{-1} - \Sigma_\ell^{-1}\bar{\Sigma}_{t,\ell-1}\Sigma_\ell^{-1}\right)W_\ell\Sigma_{\ell+1}^{\frac{1}{2}}), \\
&= \frac{1}{d}(d + \operatorname{Tr}(\Sigma_{\ell+1}^{\frac{1}{2}}W_\ell^\top\left(\Sigma_\ell^{-1} - \Sigma_\ell^{-1}\bar{\Sigma}_{t,\ell-1}\Sigma_\ell^{-1}\right)W_\ell\Sigma_{\ell+1}^{\frac{1}{2}}), \\
&\leq 1 + \frac{1}{d}\sum_{k=1}^d \lambda_1(\Sigma_{\ell+1}^{\frac{1}{2}}W_\ell^\top\left(\Sigma_\ell^{-1} - \Sigma_\ell^{-1}\bar{\Sigma}_{t,\ell-1}\Sigma_\ell^{-1}\right)W_\ell\Sigma_{\ell+1}^{\frac{1}{2}}, \\
&\leq 1 + \frac{1}{d}\sum_{k=1}^d \lambda_1(\Sigma_{\ell+1})\lambda_1(W_\ell^\top W_\ell)\lambda_1\left(\Sigma_\ell^{-1} - \Sigma_\ell^{-1}\bar{\Sigma}_{t,\ell-1}\Sigma_\ell^{-1}\right), \\
&\leq 1 + \frac{1}{d}\sum_{k=1}^d \lambda_1(\Sigma_{\ell+1})\lambda_1(W_\ell^\top W_\ell)\lambda_1\left(\Sigma_\ell^{-1}\right), \\
&\leq 1 + \frac{1}{d}\sum_{k=1}^d \frac{\sigma_{\ell+1}^2}{\sigma_\ell^2} = 1 + \frac{\sigma_{\ell+1}^2}{\sigma_\ell^2},
\end{aligned}
\tag{33}
$$

where we use the assumption that $\lambda_1(W_\ell^\top W_\ell) = 1$ **(A2)** and that $\lambda_1(\Sigma_{\ell+1}) = \sigma_{\ell+1}^2$ and $\lambda_1(\Sigma_\ell^{-1}) = 1/\sigma_\ell^2$. This is because $\Sigma_\ell = \sigma_\ell^2 I_d$ for any $\ell \in [L+1]$. Finally, plugging Eqs. (30) and (31) in Eq. (29) concludes the proof.

### C.5 PROOF OF PROPOSITION 1

We use exactly the same proof in Appendix C.4, with one change to account for the sparsity assumption **(A3)**. The change corresponds to Eq. (31). First, recall that Eq. (31) writes

$$
\sum_{t=1}^n \log(1 + \sigma^{-2}X_t^\top P_{A_t,\ell}\bar{\Sigma}_{t,\ell}P_{A_t,\ell}^\top X_t) \leq \sigma_{\text{MAX}}^{2\ell}\left(\log\det(\Sigma_{\ell+1}^{\frac{1}{2}}\bar{\Sigma}_{n+1,\ell}^{-1}\Sigma_{\ell+1}^{\frac{1}{2}})\right),
$$

where

$$
\begin{aligned}
\Sigma_{\ell+1}^{\frac{1}{2}}\bar{\Sigma}_{n+1,\ell}^{-1}\Sigma_{\ell+1}^{\frac{1}{2}} &= I_d + \Sigma_{\ell+1}^{\frac{1}{2}}W_\ell^\top\left(\Sigma_\ell^{-1} - \Sigma_\ell^{-1}\bar{\Sigma}_{t,\ell-1}\Sigma_\ell^{-1}\right)W_\ell\Sigma_{\ell+1}^{\frac{1}{2}}, \\
&= I_d + \sigma_{\ell+1}^2 W_\ell^\top\left(\Sigma_\ell^{-1} - \Sigma_\ell^{-1}\bar{\Sigma}_{t,\ell-1}\Sigma_\ell^{-1}\right)W_\ell,
\end{aligned}
\tag{34}
$$

where the second equality follows from the assumption that $\Sigma_{\ell+1} = \sigma_{\ell+1}^2 I_d$. But notice that in our assumption, **(A3)**, we assume that $W_\ell = (\bar{W}_\ell, 0_{d,d-d_\ell})$, where $\bar{W}_\ell \in \mathbb{R}^{d \times d_\ell}$ for any $\ell \in [L]$. Therefore, we have that for any $d \times d$ matrix $B \in \mathbb{R}^{dd \times d}$, the following holds, $W_\ell^\top B W_\ell = \begin{pmatrix} \bar{W}_\ell^\top B \bar{W}_\ell & 0_{d_\ell,d-d_\ell} \\ 0_{d-d_\ell,d_\ell} & 0_{d-d_\ell,d-d_\ell} \end{pmatrix}$. In particular, we have that

$$
W_\ell^\top\left(\Sigma_\ell^{-1} - \Sigma_\ell^{-1}\bar{\Sigma}_{t,\ell-1}\Sigma_\ell^{-1}\right)W_\ell = \begin{pmatrix} \bar{W}_\ell^\top\left(\Sigma_\ell^{-1} - \Sigma_\ell^{-1}\bar{\Sigma}_{t,\ell-1}\Sigma_\ell^{-1}\right)\bar{W}_\ell & 0_{d_\ell,d-d_\ell} \\ 0_{d-d_\ell,d_\ell} & 0_{d-d_\ell,d-d_\ell} \end{pmatrix}.
\tag{35}
$$

Therefore, plugging this in Eq. (34) yields that

$$
\Sigma_{\ell+1}^{\frac{1}{2}}\bar{\Sigma}_{n+1,\ell}^{-1}\Sigma_{\ell+1}^{\frac{1}{2}} = \begin{pmatrix} I_{d_\ell} + \sigma_{\ell+1}^2\bar{W}_\ell^\top\left(\Sigma_\ell^{-1} - \Sigma_\ell^{-1}\bar{\Sigma}_{t,\ell-1}\Sigma_\ell^{-1}\right)\bar{W}_\ell & 0_{d_\ell,d-d_\ell} \\ 0_{d-d_\ell,d_\ell} & I_{d-d_\ell} \end{pmatrix}.
\tag{36}
$$

As a result, $\det(\Sigma_{\ell+1}^{\frac{1}{2}}\bar{\Sigma}_{n+1,\ell}^{-1}\Sigma_{\ell+1}^{\frac{1}{2}}) = \det(I_{d_\ell} + \sigma_{\ell+1}^2\bar{W}_\ell^\top\left(\Sigma_\ell^{-1} - \Sigma_\ell^{-1}\bar{\Sigma}_{t,\ell-1}\Sigma_\ell^{-1}\right)\bar{W}_\ell)$. This allows us to move the problem from a $d$-dimensional one to a $d_\ell$-dimensional one. Then we use the inequality

of arithmetic and geometric means and get that

$$
\sum_{t=1}^{n} \log(1 + \sigma^{-2} X_t^\top \mathrm{P}_{A_t,\ell} \bar{\Sigma}_{t,\ell} \mathrm{P}_{A_t,\ell}^\top X_t) \le \sigma_{\mathrm{MAX}}^{2\ell} \left( \log \det(\Sigma_{\ell+1}^{\frac{1}{2}} \bar{\Sigma}_{n+1,\ell}^{-1} \Sigma_{\ell+1}^{\frac{1}{2}}) \right),
$$

$$
= \sigma_{\mathrm{MAX}}^{2\ell} \log \det(I_{d_\ell} + \sigma_{\ell+1}^2 \bar{\mathrm{W}}_\ell^\top \left( \Sigma_\ell^{-1} - \Sigma_\ell^{-1} \bar{\Sigma}_{t,\ell-1} \Sigma_\ell^{-1} \right) \bar{\mathrm{W}}_\ell),
$$

$$
\le d_\ell \sigma_{\mathrm{MAX}}^{2\ell} \log \left( \frac{1}{d_\ell} \mathrm{Tr}(I_{d_\ell} + \sigma_{\ell+1}^2 \bar{\mathrm{W}}_\ell^\top \left( \Sigma_\ell^{-1} - \Sigma_\ell^{-1} \bar{\Sigma}_{t,\ell-1} \Sigma_\ell^{-1} \right) \bar{\mathrm{W}}_\ell) \right),
$$

$$
\le d_\ell \sigma_{\mathrm{MAX}}^{2\ell} \log \left( 1 + \frac{\sigma_{\ell+1}^2}{\sigma_\ell^2} \right). \tag{37}
$$

To get the last inequality, we use derivations similar to the ones we used in Eq. (33). Finally, the desired result in obtained by replacing Eq. (31) by Eq. (37) in the previous proof in Appendix C.4.

## D  ADDITIONAL EXPERIMENT

**Regret scaling with $K$, $L$, $d$.** In this experiment, we aim to empirically verify the relationships outlined in Theorem 1 between the regret of dTS and several key factors: the number of actions $K$, the context dimension $d$, and the diffusion depth $L$. We maintain the same experimental setup with linear rewards, for which we have derived a Bayes regret. In Fig. 3, we plot the regret of LindTS across varying values of these parameters: $K \in \{10, 100, 500, 1000\}$, $d \in \{5, 10, 15, 20\}$, and $L \in \{2, 4, 5, 6\}$. As anticipated and aligned with our theory, the empirical regret increases as the values of $K$, $d$, or $L$ grow. This trend arises because larger values of $K$, $d$, or $L$ result in problem instances that are more challenging to learn, consequently leading to higher regret. Interestingly, the empirical regret of LindTS increases as the number of actions increases, which is consistent with the regret bound outlined in Theorem 1. This observation may appear counterintuitive, as one might expect the regret to depend solely on the diffusion depth $L$ in our setting. However, as discussed in Section 4, this behavior is caused by the fact that action parameters are not deterministic given the latent parameters.

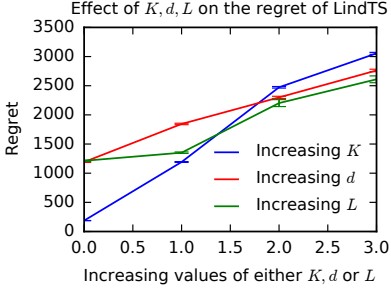

Figure 3: Regret of LindTS w.r.t. $K, d, L$.

