# OpenReview forum: "Linear diffusion models meet contextual bandits with large action spaces"
_ICLR.cc/2024/Conference — Submitted to ICLR 2024_

### Official Review · Reviewer_S8eU · 2023-10-29

**Soundness:** 2 fair
**Presentation:** 3 good
**Contribution:** 1 poor
**Rating:** 3
**Confidence:** 4

**Summary:**

The paper extends prior work on diffusion model-based TS with MAB to contextual bandits, and specifically, with a so-called linear diffusion model for linear contextual bandits. The corresponding TS algorithm is derived and analyzed, and numerical results are presented to show advantages over a few baselines.

**Strengths:**

The paper writing is clear and easy to follow

**Weaknesses:**

The main contribution of this paper (linear diffusion model for linear contextual bandits) is exactly equivalent to prior works, which makes the novelty and contribution very limited. See below.

**Questions:**

It is important to clarify the distinction of linear diffusion models with the "correct" equivalent form of HierTS and (1). Indeed, in (4) or (9), with W'sand \Sigma's as known, it is clear that the multiple layer of linear Gaussian transformation is just equivalent to one layer of linear Gaussian prior, and hence effectively equivalent to HierTS or (1). I notice the authors tried to clarify this point in (17), but in wrong way (which I hope is not on purpose) - instead of keeping the first layer and marginalizing the other latent layers, you should actually marginalizing all layers in the prior. The equivalent to linear bandits with linear Gaussian prior (and ignoring estimation error of the diffusion model) is exactly the reason why you can get a theoretical guarantee).


[1] Aouali I, Kveton B, Katariya S. Mixed-effect thompson sampling[C]//International Conference on Artificial Intelligence and Statistics. PMLR, 2023: 2087-2115.

---

> ### Author Response · Authors · 2023-11-17
> **Part 1**
>
> We would like to thank you for your feedback and valuable time. We provide our response below.
>
> __dTS and Gaussianity__
>
> While our posterior derivations and Lemma 2 affirm the Gaussian nature of $\theta_i|H_t$ , the structure of our proof is different from standard Gaussian Thompson Sampling (TS). In fact, using standard Gaussian TS results would lead to a Bayes regret bound that doesn't fully capture the structure of our problem (correct dependencies on $L$, $\sigma_\ell$, and more importantly sparsity dimensions $d_\ell$). We acknowledged in the paper that the Gaussian form of the posterior leads to Lemma 1 which is borrowed from existing works. However,
>  all the other elements, including Lemmas 2 and 3, Theorem 1, and Proposition 1, introduce novel concepts specifically designed for our model's unique framework, and are not standard. More details about the novelty of these results will be provided in the next part of our response. Now it is important to note that the primary objective of our study is to theoretically elucidate why diffusion models are effective priors for TS, complemented by experimental evidence supporting these theoretical insights. dTS was designed for general diffusion models, our current analysis focuses on the linear case. This analysis distinctly highlights the differences between dTS and standard TS, potentially motivating future explorations into analyzing the non-linear version of dTS.
>
> __Comparison to HierTS__
>
> Our approach to marginalizing latent parameters aligns with common practices (e.g., Tree-HierTS (Section 6 in [1])), to ensure a fair computational efficiency comparison. This is analogous to how two-level HierTS is typically compared to Independent-TS, which models each action parameter independently, rather than Joint-TS, which models the joint distribution but is computationally more intensive. For example, under specific choices of mixing matrices in dTS such that the coordinates of $\psi_1$ independent given $\psi_2, H_t$, our computational efficiency scales with $d_1$ rather than $d_1^3$ due to the feasibility of independent coordinate sampling. In contrast, two-level HierTS, with traditional marginalization, would scale with $d_1^3$ as conditional independence is lost after marginalizing $\psi_2$.
>
> Beyond these comparisons, dTS is designed for more general cases (Section 3), where marginalization to two-level HierTS is not always possible. Additionally, direct application of the theory of two-level HierTS does not replicate our results, even in Gaussian cases. Our model deviates by using multiple levels, wherein a function $f_1(\psi_1)$, or $W_1 \psi_1$ in linear scenarios, is used as the mean for action parameters, unlike two-level HierTS's assumption of Gaussian distributions centered at a common latent parameter (which corresponds to $W_1=I_d$). In particular, the use of linear functions $W_1 \psi_1$ allows the introduction of the sparsity assumption, which is a critical aspect not possible in standard HierTS. While Mixed-Effect TS also proposes a two-level hierarchy with Gaussian action parameters based on a linear mixture of latent parameters, our model extends beyond this, treating it as a special case with $L=1$ and specific mixing matrices. More technical novelties of our work are detailed subsequently.
>
> [1] Hong, Joey, et al. "Deep Hierarchy in Bandits." arXiv preprint arXiv:2202.01454 (2022).

---

> > ### Author Response · Authors · 2023-11-17
> > **Part 2**
> >
> > __Technical Contributions__
> >
> > As we mentioned in Part 1, the fact that the posterior distribution of $\theta_i | H_t$ is multivariate Gaussian only allow us to get Lemma 1, all other results are novel. We have included a detailed discussion of the technical challenges and contributions of their proofs in Appendix C.1. Below is a more succinct summary of these points.
> >
> > - __Lemma 2.__ In dTS, sampling is done hierarchically, and thus the marginal posterior distribution of $\theta_i | H_t$ is not explicitly defined since we only use the conditional posterior distribution of $\theta_i | H_t, \psi_1$. Our first contribution was deriving the covariance of $\theta_i | H_t$ using total covariance decomposition combined with an induction argument, as our posteriors in Section 3.1 were derived recursively. This differs from the standard Bayes regret, where the posterior distribution of $\theta_i | H_t$ is predetermined due to the absence of latent parameters. Note that the total covariance decomposition was originally used in HierTS for multi-task linear bandits with a single latent parameter. Here, we extend it to contextual bandits with multiple, successive levels of latent parameters. Furthermore, our proof diverges from HierTS by incorporating linear functions ($W_\ell\psi_\ell$), as opposed to HierTS's reliance on the more straightforward identity function $\psi_\ell$.
> > - __Lemma 3.__ Standard Bayes regret analysis quantifies the posterior information gain from actions, particularly by examining the increase in posterior precision of the taken action $A_t$. Our proof, however, requires considering the information gain for each latent parameter, since they are also learned. Then, our contribution lies in using recursive formulas in Section 3.1 to propagate the posterior information gain of the taken action $A_t$ to all latent parameters within the range $\ell \in [L]$. This is not only novel but also technically challenging, as it combines recursive formulas, induction arguments, the Sherman–Morrison formula, and properties of rank-1 matrices. This enables a nuanced quantification of how the information gain from a particular action can impact the learning of latent parameters.
> > - __Proposition 1.__ Proposition 1 enhances Theorem 1's Bayes regret analysis through the sparsity assumption __(A3)__, which is grounded in the intuitive idea that sparsity leads to fewer parameters to learn, and hence smaller regret. This assumption refines regret to depend on the reduced dimensions $d_\ell \leq d$, instead of $d$. We adapt Theorem 1's proof by partitioning the covariance matrix of each latent parameter $\psi_\ell$ into two blocks: a $d_\ell \times d_\ell$ block for learnable parameters and a $(d-d_\ell) \times (d-d_\ell)$ block for non-learnable ones. This leads to the important finding that the final regret depends only on the learnable $d_\ell$ parameter, leading to tighter regret and highlighting the impact of the sparsity assumption.

---

> > > ### Comment · Reviewer_S8eU · 2023-11-20
> > >
> > > Thank you for your reply. I agree considering sparsity in multi-layer prior has benefits in theory and computation), and non-linear case is different from Hier-TS. Though the former is actually a very strong assumption and the novelty in derivations is limited, and little efforts are actual spent on the latter. We may have different assessment but these are not what I want to focus on.
> > >
> > > Consider the linear-linear-Guassian case which is your main focus and where novelty mainly comes from.
> > >
> > > I think we agree both papers study how to model the prior of $\theta_{\*,i}$.
> > >
> > > You claim in (17) that Hier-TS should be
> > >
> > > $\psi_{*,L} \sim N(0, \Sigma_{L+1}) $   (meta-prior)
> > >
> > > $\theta_{\*, i} | \psi_{\*,L} \sim N(\psi_{\*,L}, \Omega) $ (prior)
> > >
> > > This marginalization increases the prior variance.
> > >
> > > Another marginalization is
> > >
> > > $\psi_{*,1} \sim N(0, \Omega')$  (meta-prior)
> > >
> > > $\theta_{\*, i} | \psi_{\*,L} \sim N(W_1 \psi_{\*,1}, \Omega). $ (prior)
> > >
> > > I have a few questions. Put sparsity aside (consider all matrics are dense).
> > >
> > > 1. Is there any theoretical difference between the two marginalization, when applying Hier-TS's result?
> > > 2. What is the difference between your method/result with the second marginalization of Hier-TS in terms of either theory or computation (I understand you do computation hierarchically; but is there any gain without sparsity assumption)? You mentioned in your reply that Hier-TS assumes $W_1$ is an identity matrix, while it is quite unclear to me why that is a drawback. Indeed, your method/result looks **identical** with this formulation of Hier-TS.
> > >
> > > I am happy to consider increasing the score if the authors are willing to discuss these relationship, both here and also more openly in the paper.

---

> > > > ### Author Response · Authors · 2023-11-21
> > > > **Follow-up Part (I)**
> > > >
> > > > We are really grateful for your valuable feedback and help in refining our paper, particularly our discussion. We have enhanced Section 4.1 based on your suggestions. Below, we address your questions and have also omitted the $*$ subscript due to rendering issues.
> > > >
> > > > 1. Yes, there is a difference between the two marginalizations. To provide clarity, let's begin with the foundational assumption made by HierTS [1]. Precisely, HierTS assumes that
> > > > $$\psi \sim N(0, \Omega)$$
> > > > $$\theta_i | \psi \sim N(\psi, \Sigma),$$
> > > > for some covariance matrices $\Omega$ and $\Sigma$. In particular, HierTS [1] centers action parameters around a singular latent parameter $\psi$. Interestingly, linear diffusion models, which employ linear functions $W_\ell \psi_\ell$, can be adapted to fit exactly into the above 2-level hierarchy (with centered action parameters). Thus, we will propose a slight modification to the marginalization you suggested, but our marginalization remains equivalent to yours and is only used to be able to apply HierTS theorems. Indeed, there are two distinct marginalization strategies available, with the first marginalization strategy is
> > > > $$\psi_L \sim N(0, \sigma_{L+1}^2 B_L B_L^\top)$$
> > > > $$\theta_i | \psi_L  \sim N( \psi_L , \Omega_1) \qquad (1)$$
> > > > where $\Omega_1 = \sigma_1^{2} I_d +  \sum_{\ell \in [L-1]} \sigma_{\ell+1}^2  B_\ell B_\ell^\top$ and $B_\ell = \prod_{k \in [\ell]}W_k$. We refer to HierTS when employing the first marginalization strategy (Eq. (1)) as HierTS-1. The second marginalization strategy yields,
> > > > $$\psi_1 \sim N(0, \Omega_2)$$
> > > > $$\theta_i \mid \psi_1  \sim N( \psi_1 ,  \sigma_1^2 I_d) \qquad (2)$$
> > > > where $\Omega_2 =  \sum_{\ell \in [L]} \sigma_{\ell+1}^2  B_\ell B_\ell^\top$. We refer to HierTS when employing the second marginalization strategy (Eq. (2)) as HierTS-2. Then, the regrets of HierTS-1 and HierTS-2 scale as (refer to the important note below for details on the derivation of these regrets)
> > > > $$\text{HierTS-1:} \qquad \tilde{O}\big(\sqrt{nd(K \sum_{\ell \in [L]}\sigma_\ell^2 + L \sigma_{L+1}^2}\big)$$
> > > > $$\text{HierTS-2:} \qquad \tilde{O}\big(\sqrt{n d (K \sigma_1^2 + \sum_{\ell \in [L]} \sigma_{\ell+1}^2 )}\big)$$
> > > > Then when $K$ and $L$ are comparable, the regrets of HierTS-1 and HierTS-2 are similar. However, in more common scenarios where $K > L$, HierTS-2 tends to outperform HierTS-1. This superiority stems from HierTS-2's strategy of putting more uncertainty to the $d$-dimensional latent parameter $\psi_1$, rather than to $K$ individual $d$-dimensional action parameters $\theta_i$ for $i \in [K]$. Additionally, HierTS-1's regret is higher because applying it requires assuming that $\theta_i$ are conditionally independent given $\psi_L$, which may not always hold true. Consequently, HierTS-2 outperforms HierTS-1.
> > > >
> > > > __Important Note.__ To clarify our regret derivations for HierTS, note that HierTS [1] was designed for multi-task linear bandits, and our discussion is based on translating HierTS's regret to the contextual bandit setting. This translation can be approached in two ways, such as applying our Theorem 1 with the marginalized covariances or using the Mixed-Effect TS [2] under the assumption of a single effect, since [2] was designed for contextual bandits. But this still needs additional care because both of these results assume covariances in the form $\Sigma=vI_d$. However, post-marginalization, the marginal covariances no longer have this form; they are dense. Fortunately, the results are still applicable to arbitrary covariance matrices. To adapt these theorems to such scenarios, we replace the scalar variances $v$ with $\lambda_1(\Sigma)$, the largest eigenvalue of the covariance $\Sigma$.
> > > >
> > > >
> > > > [1] Joey Hong, Branislav Kveton, Manzil Zaheer, and Mohammad Ghavamzadeh. Hierarchical Bayesian
> > > > bandits. In Proceedings of the 25th International Conference on Artificial Intelligence and Statistics,
> > > > 2022b.
> > > >
> > > > [2] Imad Aouali, Branislav Kveton, and Sumeet Katariya. Mixed-effect thompson sampling. In International Conference on Artificial Intelligence and Statistics, pp. 2087–2115. PMLR, 2023.

---

> > > > > ### Author Response · Authors · 2023-11-21
> > > > > **Follow-up Part (II)**
> > > > >
> > > > > 2. Regarding dTS, and __in the absence of the sparsity assumption__, HierTS-2 and dTS essentially become equivalent. Specifically, their regrets' dependency on $K$ is identical, where both methods involve multiplying $K$ by $\sigma_1^2$, leading to improved performance compared to HierTS-1. In scenarios with dense mixing matrices, where assumption __(A3)__ is not applicable, Theorem 1 indicates that dTS and HierTS-2 would demonstrate comparable levels of regret and computational efficiency. However, under the sparsity assumption and with certain mixing matrices that allow for conditional independence of $\psi_1$ coordinates given $\psi_2$, dTS acquires a great computational advantage over HierTS-2. This distinction explains why studies focusing on multi-level hierarchies typically benchmark their algorithms against two-level structures akin to HierTS-1, rather than the more competitive HierTS-2. This approach is consistent with previous works in Bayesian bandits using multi-level hierarchies, such as Tree-based priors [3], which often favor comparisons with HierTS-1. In line with this, we also compared dTS with HierTS-1 in our experiments.
> > > > >
> > > > > We hope that our clarifications have addressed your comments. We have also incorporated these explanations into Section 4.1 of our paper. Should you have any more questions, we are more than willing to engage in additional discussions. Thank you once again for your valuable feedback, which has contributed to the enhancement of our understanding of the comparison with HierTS and why prior works compared their algorithms to HierTS-1 rather than HierTS-2, which we followed in our work.
> > > > >
> > > > > [3] Hong, Joey, et al. "Deep Hierarchy in Bandits." arXiv preprint arXiv:2202.01454 (2022).

---

> ### Comment · Reviewer_S8eU · 2023-11-21
>
> Thanks for the clarification to the question of mine and and Reviewer hKWS!
>
> My take is the following:
>
> 1. The **statistical property/performance** of HierTS-2 and dTS should be identical, for any models you study (recall HierTS was also defined in a general way as you are doing, though both without much analysis). This is because the two posterior distributions are just the same (though sampling is done in different ways).
> 2. For **linear-linear-Gaussian without sparsity** (actually low-rank?) in prior: the theoretical performance is comparable (dTS is actually slightly worse with 2^l term), and the computation of dTS is also heavier.
> 3. For **linear-linear-Gaussian with sparsity** (actually low-rank?) in prior: your paper provides a tighter regret bound under this more specific structure, and improves computation by hierarchical sampling
>
> I will raise my score to 3 given the connection with prior work is now discussed more openly, but I am still inclined to rejection given the statistical property/performance (if not algorithm, with the only difference in computation) is the same with HierTS-2.
>
> Please let me know if I and/or Reviewer hKWS misunderstand any point. Thanks.

---

> > ### Author Response · Authors · 2023-11-23
> > **Thank you!**
> >
> > We are grateful for the increase in score and your guidance in refining our paper. Regarding your point about the distinction between sparsity and low-rank assumptions, diffusion models typically involve successive latent parameters of the same dimension $d$, making the assumption of low-rank latent parameters somewhat misaligned with standard diffusion model frameworks. On the other hand, the sparsity assumption that we introduced, while producing effects akin to low-rank (reducing the number of latent parameters that need to be learned), does not impose an actual low-rank structure. This respects the conventional format of diffusion models, where latent parameters maintain consistent dimensions across levels.

---

### Official Review · Reviewer_zHzJ · 2023-10-30

**Soundness:** 4 excellent
**Presentation:** 4 excellent
**Contribution:** 2 fair
**Rating:** 6
**Confidence:** 4

**Summary:**

This paper considers the Bayesian bandit problem where the priors of underlying parameters are generated by a known linear diffusion model. The authors provide a closed-form posterior distribution formula and then apply the Thompson sampling algorithm based on this formula, establishing the corresponding Bayesian regret guarantee.

**Strengths:**

1. The motivation is compelling, given the practical success of the diffusion prior in bandit problems.
2. The paper is well-written with both the problem setting and the results stated clearly.
3. The results are sound, supported by detailed proofs and simulation results.
4. The comparsion to previous algorithms TS and HierTS is insightful and reveals benefits on using the refined prior information instead of the marginalized prior.

**Weaknesses:**

1. The adoption of the diffusion prior in bandits is interesting and powerful according to previous empirical studies. However, the paper's focus on the linear diffusion prior, essentially a linear Gaussian system, simplifies the problem considerably. Although deriving the exact formula for the posterior mean and covariance, as indicated in equations (5)-(8) of this paper, requires some calculations(see also weakness 2, 3, 4), once these results are established, I don't observe too much additional difficulty compared to prior Thompson sampling papers for linear Gaussian bandits.

2. As pointed out in the first weakness, the linear diffusion considered in this paper is actually a linear Gaussian system (LGS). Several standard results from LGS theory could be directly applied to this paper to simplify the proof. For example:

(2.1). Proposition 2 in Appendix B is straightforward from equations (13.89) and (13.90) in [1] when the parameters $\Gamma = 0, A = I, z_1 = W_1 \psi_{\*,1}, V_0 = \Sigma_1, \Sigma = \sigma^2 I,$ and $C_n = \mathbf{1}(A_n = i) X_{n}^\top$ are substituted into equations (13.78)-(13.83) of [1]. Here, $\Gamma, A, z_1, V_0,C_n$ and $\Sigma$ are notations from [1], while $W_1, \psi_{\*,1}, \Sigma_1, \sigma^2, A_n,$ and $X_n$ come from the paper. It's worth noting that although [1] let $C_n \equiv C$ for the sake of simplicity, generalizing to allow varying $C_n$ is straightforward.

(2.2). Proposition 3 in Appendix B can be directly derived by using equations (2.109) and (2.110) from [1] and making particular choices of prior covariance and design matrices. This also makes Lemma 4 redundant since it only acts as an intermediary result to prove Proposition 3.



[1] Pattern Recognition and Machine Learning(2006), Christopher M. Bishop.

**Questions:**

I have one quesition on the algorithm design: Given the knowledge of {$Q_{t,\ell}$} and $P_{t,i}$, it appears that the posterior distribution of $\( \theta_i \)$ given $H_{t}$ can be computed analytically using the properties of Gaussian distributions. Why do the authors tend to firstly sample {$\psi_{t,\ell}$} instead of directly sampling $\theta_{t,i}$ from this distribution?

---

> ### Author Response · Authors · 2023-11-17
>
> Thank you very much for your positive feedback and valuable time. We address your comments and questions below. Please let us know if you have any additional questions.
>
> __I) Technical Contributions__
> We have included a detailed discussion of the technical challenges and contributions in Appendix C.1. Below is a more succinct summary of these points.
> - __Lemma 2.__ In dTS, sampling is done hierarchically, and thus the marginal posterior distribution of $\theta_i | H_t$ is not explicitly defined since we only use the conditional posterior distribution of $\theta_i | H_t, \psi_1$. Our first contribution was deriving the covariance of $\theta_i | H_t$ using total covariance decomposition combined with an induction argument, as our posteriors in Section 3.1 were derived recursively. This differs from the standard Bayes regret, where the posterior distribution of $\theta_i | H_t$ is predetermined due to the absence of latent parameters. Note that the total covariance decomposition was originally used in HierTS for multi-task linear bandits with a single latent parameter. Here, we extend it to contextual bandits with multiple, successive levels of latent parameters. Furthermore, our proof diverges from HierTS by incorporating linear functions ($W_\ell\psi_\ell$), as opposed to HierTS's reliance on the more straightforward identity function $\psi_\ell$.
> - __Lemma 3.__ Standard Bayes regret analysis quantifies the posterior information gain from actions, particularly by examining the increase in posterior precision of the taken action $A_t$. Our proof, however, requires considering the information gain for each latent parameter, since they are also learned. Then, our contribution lies in using recursive formulas in Section 3.1 to propagate the posterior information gain of the taken action $A_t$ to all latent parameters within the range $\ell \in [L]$. This is not only novel but also technically challenging, as it combines recursive formulas, induction arguments, the Sherman–Morrison formula, and properties of rank-1 matrices. This enables a nuanced quantification of how the information gain from a particular action can impact the learning of latent parameters.
> - __Proposition 1.__ Proposition 1 enhances Theorem 1's Bayes regret analysis through the sparsity assumption __(A3)__, which is grounded in the intuitive idea that sparsity leads to fewer parameters to learn, and hence smaller regret. This assumption refines regret to depend on the reduced dimensions $d_\ell \leq d$, instead of $d$. We adapt Theorem 1's proof by partitioning the covariance matrix of each latent parameter $\psi_\ell$ into two blocks: a $d_\ell \times d_\ell$ block for learnable parameters and a $(d-d_\ell) \times (d-d_\ell)$ block for non-learnable ones. This leads to the important finding that the final regret depends only on the learnable $d_\ell$ parameter, leading to tighter regret and highlighting the impact of the sparsity assumption.
>
> __II) Posterior Derivations__
>
> We are grateful for the reviewer's insights regarding the use of results from linear Gaussian systems (LGS) to streamline our proof exposition. We plan to revise our paper to incorporate these suggestions into more simplified proofs, but we need to make sure that the direct application of these suggestions is possible due to our use of level-specific covariances and mixing matrices $\Sigma_\ell$ and $W_\ell$ (they depend on $\ell$). We will promptly update our paper with these revisions. Additionally, we'd like to note that our current proofs for posterior derivations in the Gaussian case were included for completeness, following the convention in prior research on hierarchical Bayesian bandits where detailed posterior derivations are typically provided.
>
> Regarding the reviewer's question on why we opt for hierarchical sampling over marginalizing the latent parameters, the primary reason is computational efficiency. Marginalization would require calculating the mean and covariance of the marginal $\theta_i | H_t$, which entails several matrix multiplications, thereby increasing computational complexity. Additionally, marginalization becomes infeasible in the context of non-linear diffusions (Section 3), and empirical studies in prior works [1] have also demonstrated that hierarchical sampling offers more efficient approximations in non-linear diffusion models for multi-armed bandits, further justifying our choice in methodology.
>
> [1] Yu-Guan Hsieh, Shiva Prasad Kasiviswanathan, Branislav Kveton, and Patrick Blobaum. Thompson sampling with diffusion generative prior. arXiv preprint arXiv:2301.05182, 2023.

---

### Official Review · Reviewer_6rz1 · 2023-10-30

**Soundness:** 3 good
**Presentation:** 2 fair
**Contribution:** 2 fair
**Rating:** 6
**Confidence:** 3

**Summary:**

This work extends the previous study on informative prior in contextual bandits to having the prior in the form of a diffusion model, which targets to leverage the prior information to efficiently explore a large action space and make contextual bandit algorithms more practical. The classical Thompson sampling algorithm is extended with more detailed discussions performed in the case of linear diffusion models with (generalized) linear rewards. Theoretical analyses and experimental results are reported to demonstrate the effectiveness and superiority of the proposed design.

**Strengths:**

- This work nicely follows the trends of broader ML developments and incorporates the modern diffusion models into the theoretical study of bandits, which is a valid and meaningful attempt.

- The overall designs and analyses are sound based on my understanding although I have not checked all the proofs. The intuitions are also clear based on the existing works on parameterized priors in contextual bandits.

**Weaknesses:**

- Novelty. It would be nice if the authors could better illustrate how this work differs from previous works studying contextual bandits with informative priors, especially the adopted techniques. I have come across some works, e.g., [1, 2]. It seems that the essence of this work is to leverage diffusion models are priors, while diffusion models are one particular kind of graphical model. As extending the design of Thompson sampling is rather straightforward in my view, I would love to hear the authors' comments on the technical challenges in dealing with transformers as priors.

- Additionally, while the target is to handle large action space, the obtained regrets still contain $K$, i.e., the number of arms, unless $\theta$'s are fixed given $\psi_{*,1}$. I am wondering whether this meets the design goal of handling large action space.


[1]  Aouali et. al 2023, "Mixed-Effect Thompson Sampling"

[2] Wan et. al 2022 "Towards scalable and robust structured bandits: a meta-learning framework".

**Questions:**

It would be really helpful if the authors could provide additional comments on my unclear points listed in the weakness part.

---

> ### Author Response · Authors · 2023-11-17
>
> We would like to thank you very much for your positive feedback and valuable time. We split our response into two parts. Please let us know if you have any additional questions or concerns.
>
> __I) Novetly__
>
> __Algorithmic novelty.__ Hierarchical Thompson sampling has been previously explored in simpler graphical models; our work expands this to diffusion models. While adapting hierarchical sampling to multi-level graphical models is conceptually straightforward, deriving posteriors can be complex. We offer general formulas for approximations applicable in general scenarios (Section 3). Then, specifically, for linear diffusions with linear rewards  (Section 3.1), we achieve closed-form Gaussian posteriors computed recursively in an efficient way. For linear diffusions with non-linear rewards (Section 3.2), we implement an efficient Laplace approximation, leveraging the recursive formulas from Section 3.1. Both approaches demonstrate excellent empirical performance. Notably, the recursive formulas from Section 3.1 also facilitated the development of a regret bound.
>
> __Theoretical novelty.__ We have included a detailed discussion of the technical challenges and contributions in Appendix C.1. Below is a more succinct summary of these points.
>
> - __Lemma 2.__ In dTS, sampling is done hierarchically, and thus the marginal posterior distribution of $\theta_i | H_t$ is not explicitly defined since we only use the conditional posterior distribution of $\theta_i | H_t, \psi_1$. Our first contribution was deriving the covariance of $\theta_i | H_t$ using total covariance decomposition combined with an induction argument, as our posteriors in Section 3.1 were derived recursively. This differs from the standard Bayes regret, where the posterior distribution of $\theta_i | H_t$ is predetermined due to the absence of latent parameters. Note that the total covariance decomposition was originally used in HierTS for multi-task linear bandits with a single latent parameter. Here, we extend it to contextual bandits with multiple, successive levels of latent parameters. Furthermore, our proof diverges from HierTS by incorporating linear functions ($W_\ell\psi_\ell$), as opposed to HierTS's reliance on the more straightforward identity function $\psi_\ell$.
> - __Lemma 3.__ Standard Bayes regret analysis quantifies the posterior information gain from actions, particularly by examining the increase in posterior precision of the taken action $A_t$. Our proof, however, requires considering the information gain for each latent parameter, since they are also learned. Then, our contribution lies in using recursive formulas in Section 3.1 to propagate the posterior information gain of the taken action $A_t$ to all latent parameters within the range $\ell \in [L]$. This is not only novel but also technically challenging, as it combines recursive formulas, induction arguments, the Sherman–Morrison formula, and properties of rank-1 matrices. This enables a nuanced quantification of how the information gain from a particular action can impact the learning of latent parameters.
> - __Proposition 1.__ Proposition 1 enhances Theorem 1's Bayes regret analysis through the sparsity assumption __(A3)__, which is grounded in the intuitive idea that sparsity leads to fewer parameters to learn, and hence smaller regret. This assumption refines regret to depend on the reduced dimensions $d_\ell \leq d$, instead of $d$. We adapt Theorem 1's proof by partitioning the covariance matrix of each latent parameter $\psi_\ell$ into two blocks: a $d_\ell \times d_\ell$ block for learnable parameters and a $(d-d_\ell) \times (d-d_\ell)$ block for non-learnable ones. This leads to the important finding that the final regret depends only on the learnable $d_\ell$ parameter, leading to tighter regret and highlighting the impact of the sparsity assumption.

---

> ### Author Response · Authors · 2023-11-17
>
> __II) Large Action Spaces__
>
> We explain next how dTS is efficient in large action spaces using both theory and practice.
>
> - __Theory.__ Roughly speaking, the regret bound of dTS scales with $K \sigma_1^2$, contrasting with the standard $K \sum_{\ell}\sigma_\ell^2$ scaling, which proves advantageous especially when $\sigma_1$ is small, as often seen in diffusion models. In particular, the regret becomes independent of $K$ when $\sigma_1=0$. Moreover, analysis in Section 4.1 shows that the performance gap between dTS and LinTS is more pronounced in large action spaces, highlighting its suitability for such scenarios. There are some works [1,2,3] where the regret does not depend on $K$. However, our setting differs significantly from [1,2,3], and the dependency on $K$ is unavoidable in our setting when $\sigma_1>0$. Precisely, while [1,2,3] use a reward function $r(x, i) = \phi(x, i)^\top \theta_*$ with a common $\theta_*$ and a known mapping $\phi$, our model employs $r(x, i) = x^\top \theta_{*, i}$, necessitating the learning of $K$ distinct $d$-dimensional parameters. In the setting of [1,2,3] with an available mapping $\phi$, dTS’s regret would be independent of $K$. However, the practical challenge of obtaining such a mapping, which must capture complex context-action relations, makes our setting relevant. For example, in recommendation systems, a product is often associated with its separate embedding. To summarize, the dependency on $K$ is intrinsic to our specific setting rather than a limitation of dTS.
>
> - __Experiments.__ dTS enjoys notable computational and statistical efficiency, especially in large action spaces (Section 4.1). For instance, our empirical findings, showcased in Figure 2 with $K=10^4$, reveal that dTS substantially surpasses the baselines. More importantly, as $K$ increases, so does the performance gap between dTS and the baselines, underscoring dTS's enhanced scalability and effectiveness in large action spaces.
>
> [1] "Adapting to Misspecification in Contextual Bandits" by Foster et al. NeurIPS 2021
>
> [2] "Upper Counterfactual Confidence Bounds: a New Optimism Principle for Contextual Bandits" by Xu et al. 2021
>
> [3] "Contextual Bandits with Large Action Spaces: Made Practical" by Zhu et al. ICML 2022

---

### Official Review · Reviewer_hKWS · 2023-11-04

**Soundness:** 3 good
**Presentation:** 3 good
**Contribution:** 2 fair
**Rating:** 3
**Confidence:** 5

**Summary:**

To capture the correlations between arms, this paper incorporates diffusion models into contextual linear bandits. For this model, the authors propose the diffusion Thompson sampling (dTS) algorithm, utilizing diffusion models as priors. Notably, when the diffusion model is parameterized by linear functions, closed-form expressions can be obtained for the sampling procedure. Both theoretical analyses and empirical experiments confirm the superiority of dTS.

**Strengths:**

1. The use of diffusion models to capture correlations between arms in contextual linear bandits is impressive and highly intriguing. Personally, I think this contribution holds significant value for the bandit community.
2. Theoretical guarantees on Bayes regret are provided.  Section 4.1 compares dTS to other methods, clearly demonstrating its statistical and computational advantages under different scenarios.

**Weaknesses:**

1. While overall well-written and easy to follow, there are certain parts where I have specific questions. Please refer to my detailed queries below.
2. Despite the excellence of the proposed model, the theoretical contribution may be considered somewhat limited. Both the derivation of closed-form posteriors and the analysis of Bayes regret seem somewhat straightforward, given the existing works.

**Questions:**

1. Equation (2): At this point, as the specification to the linear regret model has not been made, the Bayes regret should be defined based on the mean of the reward distribution. Furthermore, in the definition of $A_{t, \*}$, neither the function $r()$ nor $\Theta_\*$ has been defined.
2. In the case that $\sigma_1=0$ ($\theta_{\*, i}$ is deterministic given $\psi_{*, 1}$), since all the arm vectors are the same, is it true that the regret is always zero?
3. Section 4.1: The rationale behind LinTS being suboptimal is evident. That is, LinTS needs to learn $K$ independent $d$-dimensional parameters, each with a considerably higher initial covariance $\Sigma$. However, the grounds for HierTS being suboptimal are not fully elucidated. A mere comparison of upper bounds of regrets is not equitable. More fundamental reasoning is imperative.
4. In Equation (16), it is beneficial to explicitly state that while $\theta_{*, i}$ for $i\in[K]$ follow the same normal distribution $\mathcal{N}(0, \Sigma)$, they are not independent.

---

> ### Author Response · Authors · 2023-11-17
>
> Thank you very much for your positive feedback and valuable time. We address your comments and questions below. Please let us know if you have any additional questions.
>
> __I) Theoretical Contributions__
>
> We have included a detailed discussion of the technical challenges and contributions in Appendix C.1. Below is a more succinct summary of these points.
>
> - __Lemma 2.__ In dTS, sampling is done hierarchically, and thus the marginal posterior distribution of $\theta_i | H_t$ is not explicitly defined since we only use the conditional posterior distribution of $\theta_i | H_t, \psi_1$. Our first contribution was deriving the covariance of $\theta_i | H_t$ using total covariance decomposition combined with an induction argument, as our posteriors in Section 3.1 were derived recursively. This differs from the standard Bayes regret, where the posterior distribution of $\theta_i | H_t$ is predetermined due to the absence of latent parameters. Note that the total covariance decomposition was originally used in HierTS for multi-task linear bandits with a single latent parameter. Here, we extend it to contextual bandits with multiple, successive levels of latent parameters. Furthermore, our proof diverges from HierTS by incorporating linear functions ($W_\ell\psi_\ell$), as opposed to HierTS's reliance on the more straightforward identity function $\psi_\ell$.
>
> - __Lemma 3.__ Standard Bayes regret analysis quantifies the posterior information gain from actions, particularly by examining the increase in posterior precision of the taken action $A_t$. Our proof, however, requires considering the information gain for each latent parameter, since they are also learned. Then, our contribution lies in using recursive formulas in Section 3.1 to propagate the posterior information gain of the taken action $A_t$ to all latent parameters within the range $\ell \in [L]$. This is not only novel but also technically challenging, as it combines recursive formulas, induction arguments, the Sherman–Morrison formula, and properties of rank-1 matrices. This enables a nuanced quantification of how the information gain from a particular action can impact the learning of latent parameters.
>
> - __Proposition 1.__ Proposition 1 enhances Theorem 1's Bayes regret analysis through the sparsity assumption __(A3)__, which is grounded in the intuitive idea that sparsity leads to fewer parameters to learn, and hence smaller regret. This assumption refines regret to depend on the reduced dimensions $d_\ell \leq d$, instead of $d$. We adapt Theorem 1's proof by partitioning the covariance matrix of each latent parameter $\psi_\ell$ into two blocks: a $d_\ell \times d_\ell$ block for learnable parameters and a $(d-d_\ell) \times (d-d_\ell)$ block for non-learnable ones. This leads to the important finding that the final regret depends only on the learnable $d_\ell$ parameter, leading to tighter regret and highlighting the impact of the sparsity assumption.
>
> __II) Other Questions__
>
> - __Typos and suggestions.__ We appreciate the reviewer pointing out these typos, and we fixed them in the revised version. To clarify, $\theta_*$ represents the concatenation of all $K$ action parameters. Moreover, $r(x, i; \theta_*)$ denotes the expected reward for action $i$ in context $x$. Also, we added that $\theta_{*, i}$ are not necessarily independent in Eq. (16).
>
> - __Regret when $\sigma_1=0$.__ When $\sigma_1=0$, the regret is not always zero as it is not gap dependent. Our Bayesian regret does not capture this similarly to frequentist problem-independent regret bounds.
>
> - __Additional Intuitions.__ The intuition behind the improved performance of dTS compared to HierTS is due to sparsity, especially when the sparsity dimensions $d_\ell$ are in decreasing order, such as $d_1>d_2>...>d_L$. Roughly speaking, rather than learning $d_1$ parameters with a high initial (marginal) variance, dTS learns $d_1$ parameters with a smaller initial variance, as the remaining initial variances are already incorporated in the learning of the other latent parameters, each characterized by its own small initial variance. This is why the sparsity assumption was introduced and it is key in the comparison with HierTS.

---

> > ### Comment · Reviewer_hKWS · 2023-11-21
> >
> > Thank you for the response. I would like to provide some feedback for your consideration:
> >
> > - Regarding the Bayes regret in Equation (2), could you provide additional details on why the regret is not zero when $\sigma_1=0$? A more thorough explanation would enhance the understanding of this aspect.
> >
> > -  I find that my initial confusion regarding the suboptimality of HierTS has been partially addressed by Reviewer S8eU. In my perspective, the models of HierTS and dTS should inherently be equivalent. While I understand that the formulation in (17) may be influenced by computational considerations, it seems unfair to compare their theoretical performances based on this. Therefore, I recommend that the authors clarify the parts related to HierTS for a more accurate understanding. The current presentation is misleading.
> >
> > I am keenly interested in the issues raised by Reviewer S8eU and **strongly** encourage the authors to engage in a discussion with Reviewer S8eU to address these concerns.

---

> > > ### Author Response · Authors · 2023-11-23
> > >
> > > We greatly appreciate your insightful feedback and additional comments. In the following, we respond to your questions, and please note that we have omitted the subscript * in our responses due to rendering challenges.
> > >
> > > __Bayes regret when $\sigma_1=0$.__ We apologize for any confusion; it appears there was a misunderstanding as we were talking about our regret bound in Theorem 1 while your question is regarding the Bayes regret itself in Eq. (2). Yes, the Bayes regret in Eq. (2) indeed becomes zero when $\sigma_1=0, and this is because all action parameters have the same mixing matrix.
> > >
> > > In contrast, if different mixing matrices were applied for each action parameter, such as $\theta_i \sim N(W_{1, i} \psi_1, \Sigma_1)$ with $W_{1, i}$ unique to each action $i$, the Bayes regret in Eq. (2) would not be zero. Our regret bound, as currently formulated, does not reflect this nuance, similar to gap-free bounds that remain non-zero even when expected rewards for all actions are identical.
> > >
> > > __Discussion with Reviewer S8eU.__ In line with our discussion addressed to Reviewer S8eU, our choice to compare dTS with HierTS using a particular marginalization approach is consistent with prior works and is motivated by computational considerations under the sparsity assumption. For a more detailed explanation, we encourage reviewing our response to Reviewer S8eU, and we improved Section 4.1 in our paper based on these discussions.
> > >
> > > Thanks a lot, your feedback helped us to improve our paper.

---

> > > > ### Comment · Reviewer_hKWS · 2023-11-23
> > > >
> > > > Thank you for your prompt reply.
> > > >
> > > > - I appreciate the clarification. I consistently refer to Bayes regret rather than the regret bound. It would be beneficial for the authors to explicitly differentiate between **regret** and **regret bound** in the manuscript.
> > > >
> > > >   Given that the Bayes regrets of both methods are zero, investigating the case where $\sigma_1 = 0$ seems to lack significance. Neither the bound of LTS nor the bound of dTS can effectively capture this scenario.
> > > >
> > > >   Additionally, in the manuscript, the authors state, "If we were to assume deterministic linearity ($\sigma_1 = 0$), our regret bound would scale with L only and we provide this example in Section 4.1. Consistently, our empirical results in Section 5 match **all** these findings." I suggest modifying the last sentence since this particular case is not explored in Section 5.
> > > >
> > > > -  Thank you for the discussion with Reviewer S8eU. The discussion and the take shared by Reviewer S8eU are very valuable for enhancing my understanding of the work!
> > > >
> > > > Based on the preceding discussion, I tend to disagree with accepting this work at this time. However, I believe that improvements can be made to enhance the clarity and presentation of the contributions.

---

> > > > > ### Author Response · Authors · 2023-11-23
> > > > > **Thank you!**
> > > > >
> > > > > We sincerely thank you for your valuable feedback to enhance our paper. In response to your suggestions, we have made the necessary changes to our manuscript.

---

### Official Review · Reviewer_sz3T · 2023-11-05

**Soundness:** 3 good
**Presentation:** 3 good
**Contribution:** 3 good
**Rating:** 6
**Confidence:** 4

**Summary:**

This paper studies contextual bandits with large action spaces. With prior specified by a diffusion model, the authors designed diffusion Thompson Sampling (dTS) algorithm. The authors theoretically analyze the performance of dTS, and empirically verify its efficacy.

After rebuttal: I have read the rebuttal and I'd like to keep my scores.

**Strengths:**

The authors provide the first theoretically analysis of Thompson Sampling under linear diffusion models, under certain assumptions. The authors also discuss the benefits from both computational and statistical aspects.

**Weaknesses:**

Since this paper is mainly theoretical, can the authors highlight the technical contribution of this paper? More specifically, can authors elaborate on the differences in analysis between the standard Thompson Sampling? It seems that the main difference comes from the sampling of the latent parameters \psi; but under linear diffusion model, posteriors of \psi can be easily calculated.

How does the authors address the problem with large action spaces? It seems that both statistical and computational complexities depends on poly(K), which can be exponentially large for linear bandits. Note that recent work [1, 2, 3] have removed the dependency in K, and thus work for large action spaces.

[1] "Adapting to Misspecification in Contextual Bandits" by Foster et al. NeurIPS 2021

[2] "Upper Counterfactual Confidence Bounds: a New Optimism Principle for Contextual Bandits" by Xu et al. 2021

[3] "Contextual Bandits with Large Action Spaces: Made Practical" by Zhu et al. ICML 2022

**Questions:**

See comments above.

---

> ### Author Response · Authors · 2023-11-17
>
> Thank you very much for your positive feedback and valuable time. We address your comments and questions below. Please let us know if you have any additional questions.
>
> __I) Technical Contributions__
>
> We have included a detailed discussion of the technical challenges and contributions in Appendix C.1. Below is a more succinct summary of these points.
>
> - __Lemma 2.__ In dTS, sampling is done hierarchically, and thus the marginal posterior distribution of $\theta_i | H_t$ is not explicitly defined since we only use the conditional posterior distribution of $\theta_i | H_t, \psi_1$. Our first contribution was deriving the covariance of $\theta_i | H_t$ using total covariance decomposition combined with an induction argument, as our posteriors in Section 3.1 were derived recursively. This differs from the standard Bayes regret, where the posterior distribution of $\theta_i | H_t$ is predetermined due to the absence of latent parameters. Note that the total covariance decomposition was originally used in HierTS for multi-task linear bandits with a single latent parameter. Here, we extend it to contextual bandits with multiple, successive levels of latent parameters. Furthermore, our proof diverges from HierTS by incorporating linear functions ($W_\ell\psi_\ell$), as opposed to HierTS's reliance on the more straightforward identity function $\psi_\ell$.
>
> - __Lemma 3.__ Standard Bayes regret analysis quantifies the posterior information gain from actions, particularly by examining the increase in posterior precision of the taken action $A_t$. Our proof, however, requires considering the information gain for each latent parameter, since they are also learned. Then, our contribution lies in using recursive formulas in Section 3.1 to propagate the posterior information gain of the taken action $A_t$ to all latent parameters within the range $\ell \in [L]$. This is not only novel but also technically challenging, as it combines recursive formulas, induction arguments, the Sherman–Morrison formula, and properties of rank-1 matrices. This enables a nuanced quantification of how the information gain from a particular action can impact the learning of latent parameters.
>
> - __Proposition 1.__ Proposition 1 enhances Theorem 1's Bayes regret analysis through the sparsity assumption __(A3)__, which is grounded in the intuitive idea that sparsity leads to fewer parameters to learn, and hence smaller regret. This assumption refines regret to depend on the reduced dimensions $d_\ell \leq d$, instead of $d$. We adapt Theorem 1's proof by partitioning the covariance matrix of each latent parameter $\psi_\ell$ into two blocks: a $d_\ell \times d_\ell$ block for learnable parameters and a $(d-d_\ell) \times (d-d_\ell)$ block for non-learnable ones. This leads to the important finding that the final regret depends only on the learnable $d_\ell$ parameter, leading to tighter regret and highlighting the impact of the sparsity assumption.
>
> __II) Large Action Spaces__
>
> We thank the reviewer for the references, they were added to Appendix A. We explain next how dTS is efficient in large action spaces using both theory and practice, but more details can be found in Appendix A.
>
> - __Theory.__ Roughly speaking, the regret bound of dTS scales with $K \sigma_1^2$, contrasting with the standard $K \sum_{\ell}\sigma_\ell^2$ scaling, which proves advantageous especially when $\sigma_1$ is small, as often seen in diffusion models. In particular, the regret becomes independent of $K$ when $\sigma_1=0$. Moreover, analysis in Section 4.1 shows that the performance gap between dTS and LinTS is more pronounced in large action spaces, highlighting its suitability for such scenarios.  Furthermore, our setting differs significantly from [1,2,3], and the dependency on $K$ is unavoidable in our setting when $\sigma_1>0$. Precisely, while [1,2,3] use a reward function $r(x, i) = \phi(x, i)^\top \theta_*$ with a common $\theta_*$ and a known mapping $\phi$, our model employs $r(x, i) = x^\top \theta_{*, i}$, necessitating the learning of $K$ distinct $d$-dimensional parameters. In the setting of [1,2,3] with an available mapping $\phi$, dTS’s regret would be independent of $K$. However, the practical challenge of obtaining such a mapping, which must capture complex context-action relations, makes our setting relevant. For example, recommendation systems, each product is associated with its embedding. To summarize, the dependency on $K$ is intrinsic to our specific setting rather than a limitation of dTS.
>
> - __Experiments.__ dTS enjoys notable computational and statistical efficiency, especially in large action spaces (Section 4.1). For instance, our empirical findings, showcased in Figure 2 with $K=10^4$, reveal that dTS substantially surpasses the baselines. More importantly, as $K$ increases, so does the performance gap between dTS and the baselines, underscoring dTS's enhanced scalability and effectiveness in large action spaces.

---

### Meta-Review · Area_Chair_rdiB · 2023-12-04

**Metareview:**

Reviewers S8eU and hKWS engaged in a lively and constructive discussion with the authors, pointing out some key issues that have to be addressed for the paper to be accepted. In particular, the authors are encouraged to demonstrate that the statistical property and performance of HierTS-2 and dTS are identical.

**Justification For Why Not Higher Score:**

There are several deficiencies in the current version of the paper that the authors have to fix before the paper is ready for acceptance.

**Justification For Why Not Lower Score:**

The score is already quite low.

---

### Decision · Program_Chairs · 2024-01-16

Reject